# Metabolomic profiling of rare cell populations isolated by flow cytometry from tissues

Andrew W DeVilbiss[1], Zhiyu Zhao[1], Misty S Martin-Sandoval[1], Jessalyn M Ubellacker[1], Alpaslan Tasdogan[1], Michalis Agathocleous[1], Thomas P Mathews[1,2]*, Sean J Morrison[1,2]*

[1]Children's Research Institute and Department of Pediatrics, University of Texas Southwestern Medical Center, Dallas, United States; [2]Howard Hughes Medical Institute, University of Texas Southwestern Medical Center, Dallas, United States

**Abstract** Little is known about the metabolic regulation of rare cell populations because most metabolites are hard to detect in small numbers of cells. We previously described a method for metabolomic profiling of flow cytometrically isolated hematopoietic stem cells (HSCs) that detects 60 metabolites in 10,000 cells (Agathocleous et al., 2017). Here we describe a new method involving hydrophilic liquid interaction chromatography and high-sensitivity orbitrap mass spectrometry that detected 160 metabolites in 10,000 HSCs, including many more glycolytic and lipid intermediates. We improved chromatographic separation, increased mass resolution, minimized ion suppression, and eliminated sample drying. Most metabolite levels did not significantly change during cell isolation. Mouse HSCs exhibited increased glycerophospholipids relative to bone marrow cells and methotrexate treatment altered purine biosynthesis. Circulating human melanoma cells were depleted for purine intermediates relative to subcutaneous tumors, suggesting decreased purine synthesis during metastasis. These methods facilitate the routine metabolomic analysis of rare cells from tissues.

*For correspondence: thomas.mathews@ UTSouthwestern.edu (TPM); sean.morrison@utsouthwestern. edu (SJM)

## Introduction

Metabolomics is typically performed using millions of cells, often using cultured cells, whole tissues, or tumor specimens (*Jang et al., 2018*). This measures average metabolite levels across the cells in a specimen but is blind to metabolic differences among cells in the same sample. As a result, we have limited insights into metabolic heterogeneity among cells within tissues or tumors (*Kim and DeBerardinis, 2019*; *Muir et al., 2018*). This limitation is particularly apparent when considering rare cells, such as stem cells or circulating cancer cells, that may be metabolically different from other cells. The difficulty in performing metabolomics on small numbers of these cells is compounded by the need to purify them from tissues, introducing additional technical challenges for metabolic analysis (*Binek et al., 2019*; *Llufrio et al., 2018*; *Lau et al., 2020*).

It is extremely difficult to isolate a million cells from a rare cell population by flow cytometry. One study isolated over 1 million CD34-Flt3-Lineage-Sca-1+c-kit+ hematopoietic stem cells (HSCs) by flow cytometry but pooled bone marrow cells from 120 mice to do it, precluding the analysis of multiple replicates or routine application of this approach (*Takubo et al., 2013*). Metabolomics has also been performed on hundreds of thousands of flow cytometrically isolated Lineage-Sca-1+c-kit+ (LSK) cells (*Naka et al., 2015*; *Karigane et al., 2016*), a more heterogeneous population of hematopoietic stem and progenitor cells. Since only a small minority of these cells are HSCs, this strategy provides limited insights into metabolite levels in HSCs themselves. Others have characterized the phenotypes of mutant mice or metabolism in cultured hematopoietic stem and progenitor cells

(*Simsek et al., 2010*; *Ito et al., 2012*; *Ito et al., 2016*; *Ito et al., 2019*; *Wang et al., 2014*; *Ansó et al., 2017*). However, it remains difficult to routinely compare metabolite levels between HSCs and other hematopoietic progenitors.

Metabolites have been profiled in single cells (*Evers et al., 2019*; *Comi et al., 2017*). However, these studies often use very large cells like Xenopus eggs (*Onjiko et al., 2015*) or Aplysia neurons (*Nemes et al., 2012*). Other single cell analyses have focused on small numbers of metabolites or specific subsets of metabolites (*Luo and Li, 2017*). Single cell metabolomics methods often involve mass spectrometry without chromatographic separation, making it more difficult to identify the detected species (*Duncan et al., 2019*; *Ali et al., 2019*).

We recently described a method for metabolomic analysis of highly purified, flow cytometrically isolated CD150$^+$CD48$^-$LSK HSCs that detected approximately 60 metabolites in 10,000 cells (*Agathocleous et al., 2017*). Cells were kept cold during the entire purification process and sorted directly into 80% methanol to immediately quench enzymatic activity and extract metabolites. This method revealed that HSCs take up more ascorbate than other hematopoietic cells and depend upon ascorbate for epigenetic regulation and leukemia suppression, though coverage of many metabolic pathways was limited.

The challenge of performing metabolomic analysis in rare cells is not limited to stem cells as illustrated by a paucity of information about the metabolic state of circulating cancer cells. Many tumors spontaneously shed cancer cells into the blood (*Micalizzi et al., 2017*) but these cells are extremely rare, limiting the amount of material for analysis. We have developed methods for the flow cytometric isolation and characterization of circulating human melanoma cells from the blood of xenografted mice (*Piskounova et al., 2015*; *Tasdogan et al., 2020*). These cells undergo reversible metabolic changes during metastasis to survive oxidative stress, but these changes are just beginning to be characterized. Mass spectrometric analysis of single circulating cancer cells from the blood of patients revealed metabolites that differed among various kinds of cancer cells (*Hiyama et al., 2015*; *Abouleila et al., 2019*). Fluorescent probes have also been used to characterize metabolism in circulating cancer cells (*Li et al., 2019*).

Here we present a new method for the metabolomic analysis of rare stem cell and cancer cell populations isolated by flow cytometry. We have increased the number of metabolites we can detect in 10,000 HSCs to approximately 160. Using this method, the levels of most metabolites did not significantly change during cell preparation and sorting.

## Results

### Chromatography and mass spectrometry

In order to significantly increase the numbers of metabolites we detected in small numbers of flow cytometrically isolated cells, we re-examined the chromatography and mass spectrometry approaches we used. A key limitation is discriminating the low levels of metabolites present in small numbers of cells from background signals. Background reflects contamination from various sources as well as the co-association of salts with organic compounds in mass spectrometers to generate organic salt clusters (matrix ions) that obscure the detection of metabolites. We reasoned we could improve the signal to noise ratio in low abundance samples and reduce interference by matrix ions by transitioning to a mass spectrometer with higher mass resolving power.

We chose a Q-Exactive HF-X hybrid quadrupole-orbitrap mass spectrometer (ThermoScientific) because it offers four advantages over the triple-quadrupole mass spectrometer used in our previous method (*Agathocleous et al., 2017*). First, whereas the triple-quadrupole instrument acquires data for a predetermined number of metabolites, the orbitrap instrument captures spectra for the full mass range (80–1200 Daltons) with each scan, greatly increasing the number of metabolites detected. Second, orbitrap mass analyzers have higher mass resolving power and higher mass accuracy, increasing the ability to discriminate relevant analytes from background ions. Third, through untargeted acquisition of product ion spectra, orbitrap instruments enable the comparison of spectra from experimental samples with annotated spectrum libraries for high-confidence identification of metabolites. Finally, compared to other orbitrap models, the HF-X front end optics increase the number of ions that can pass into the mass spectrometer, boosting the signal from low abundance analytes.

We also wondered if a hydrophilic interaction liquid chromatography (HILIC) system would improve the separation of polar metabolites as compared to the reverse phase chromatography method in our original study (*Agathocleous et al., 2017*). To test this, we extracted metabolites from $5 \times 10^6$ mouse whole bone marrow (WBM) cells in 500 µl of 80% methanol, dried the extracts in a vacuum concentrator, and reconstituted in water for reverse phase chromatography or 80% methanol for HILIC. Polar analytes eluted from the reverse phase column between 3 and 5 min, and from the HILIC column between 2 and 15 min, indicating that HILIC improved polar metabolite separation (data not shown).

When we ran high abundance samples either by reverse phase or HILIC we identified hundreds of metabolites by spectral database matching and manual peak review (data not shown). However, the improved metabolite separation and peak quality we observed with HILIC yielded more high confidence identifications of metabolites via spectral database matching alone (*Figure 1A*). HILIC also enabled the detection of early-eluting lipid metabolites, which were not detected using reverse phase chromatography. They were well-resolved chromatographically (*Figure 1—figure supplement 1A–D*) and were detected within the linear range of the mass spectrometer (*Figure 1—figure supplement 1E–H*). Finally, HILIC eliminated the requirement for sample drying, which can alter the levels of certain metabolites and increase contamination (*Lu et al., 2017*). We thus selected HILIC for further method development.

We also fundamentally changed our approach to data analysis. To determine which metabolites were detected in low abundance samples we created a list of metabolites with known masses and chromatographic retention times from the analysis of high abundance samples. We first used unbiased metabolite identification software (Compound Discoverer) to compare experimentally observed mass spectra with annotated spectrum libraries to identify 571 metabolites. We confirmed the identities of each metabolite in the library by reviewing the MS2 spectra for each metabolite. We confirmed the retention times and mass spectra for over 450 metabolites in the library by running chemical standards. This library was used to determine how many metabolites were detected in low abundance samples by manually analyzing chromatographic peaks derived from extracts of 100,000 WBM cells. This resulted in a low abundance library containing 283 detectable metabolites that was used for manual metabolite quantitation in low abundance samples. This manual approach was more time consuming but more accurate than relying upon automated peak-calling algorithms, which often failed to accurately integrate LC-MS peaks from low-abundance samples.

## Reducing sources of contamination

Background signals arose from the staining medium in which we suspended the cells, the flow cytometer sheath fluid, the solvent we used to extract metabolites from the sorted cells, and the drying and reconstitution of samples prior to liquid chromatography/mass spectrometry (LC-MS) (*Agathocleous et al., 2017*). While these very low levels of background would be negligible when analyzing high abundance samples, they did interfere with the ability to detect some metabolites in low abundance samples.

When performing reverse phase separation, metabolites were extracted using 80% methanol and then dried in a vacuum concentrator so they could be reconstituted in water for chromatography (*Agathocleous et al., 2017*). Transitioning to HILIC made it possible to directly inject organic solvents into the column, without drying and reconstituting in water. To test if contamination was reduced by not drying in a vacuum concentrator, we sorted droplets of sheath fluid with no cells in volumes equivalent to that required to sort 10,000 cells and processed the samples side-by-side in three ways. Some samples were dried in a standard vacuum concentrator, then reconstituted in 80% methanol, and injected into the HILIC column. Some samples were dried in a new vacuum concentrator housed in a HEPA-filtered PCR hood to minimize contamination from the air, then reconstituted in 80% methanol and injected into the HILIC column. The remaining samples were sorted into 80% methanol and injected directly into the HILIC column without drying. The highest level of background contamination was in the samples dried in the standard vacuum concentrator (*Figure 1B*). The lowest background was in the samples injected into the column without drying, suggesting drying increased contamination.

To test if we could detect more metabolites above background in low abundance samples if we did not dry and reconstitute, we sorted samples of 10,000 WBM cells, along with sheath fluid negative controls, and processed the samples side-by-side either with drying in a standard vacuum

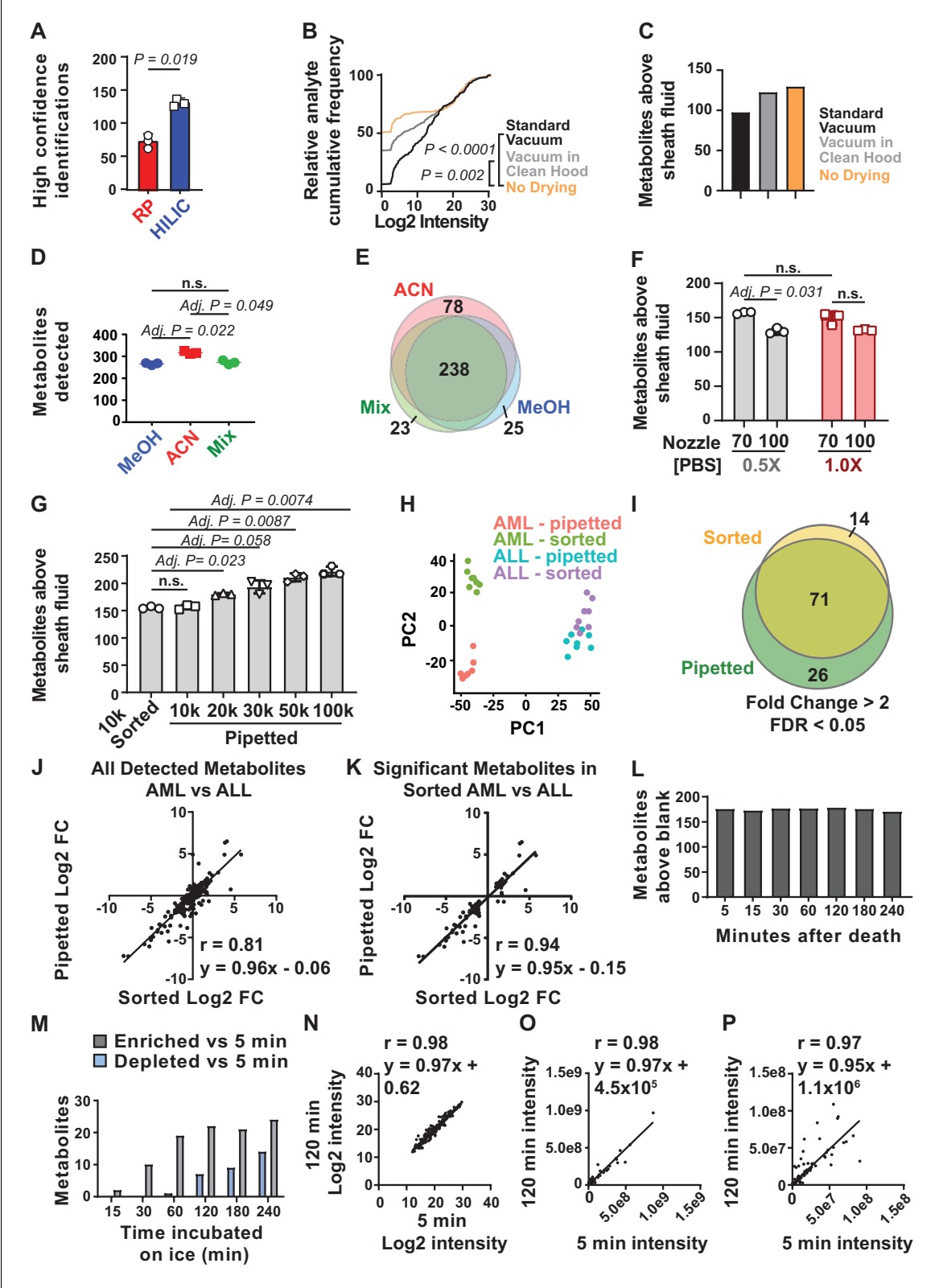

**Figure 1.** Sample processing and chromatography parameters. (**A**) The number of metabolites identified with high confidence spectral database matching in whole bone marrow (WBM) samples after HILIC or reverse phase chromatography (n = 3 replicates per group from one experiment). (**B**) Average peak intensities in sheath fluid background samples after drying with a standard vacuum concentrator, a vacuum concentrator in a positive pressure HEPA-filtered clean hood, or with no drying (n = 5 replicates per treatment from one experiment). (**C**) Number of metabolites significantly

*Figure 1 continued on next page*

*Figure 1 continued*

above sheath fluid background in 10,000 sorted WBM cells after drying with a standard vacuum concentrator, a vacuum concentrator in a clean hood, or with no drying (n = 5 replicates per treatment from one experiment). The threshold for statistical significance relative to background or other samples was always set at fold change >2 and false discovery rate (FDR) < 0.05, unless otherwise indicated. (D) Metabolites detected in 100,000 WBM cells extracted with 80% acetonitrile in water (ACN), 80% methanol in water (MeOH), or 40% ACN plus 40% MeOH in water (Mix) (n = 3 replicates per treatment from one experiment). (E) Overlap in metabolites detected with each extraction solvent (n = 3 replicates per treatment from one experiment). (F) Number of metabolites significantly above background in 10,000 WBM cells sorted using a 70 or 100 μm nozzle, and 0.5× or 1.0× phosphate buffered saline (PBS) sheath fluid (n = 5 replicates per treatment in each of three independent experiments; the metabolites that significantly differed between 0.5× PBS versus 1× PBS sheath fluid are listed in *Figure 1—source data 2*, Supplementary Table 1). (G) Number of metabolites significantly above background in 10,000 sorted WBM cells or 10,000–100,000 pipetted WBM cells (n = 5 replicates per treatment in each of three independent experiments). (H–K): (H) Principal component analysis of 10,000 sorted or pipetted HNT-34 AML (AML) cells or DND-41 T-ALL (ALL) cells (one experiment with n = 8 replicates per treatment; the metabolites that significantly differed between sorted and pipetted AML cells and between sorted and pipetted ALL cells are shown in *Figure 1—source data 3*, Supplementary Table 2 and *Figure 1—source data 4*, Supplementary Table 3). *Figure 1—source data 8* shows the raw metabolomics data for the comparison of AML to ALL cells. (I) Metabolites that significantly changed between AML and ALL cells in sorted versus pipetted samples (listed in *Figure 1—source data 5*, Supplementary Table 4). (J, K) Correlation between log2 fold changes (in AML versus ALL cells) in sorted versus pipetted samples for all detected metabolites (J) and metabolites that significantly differed between sorted AML and ALL cells (K). (L–P): (L) Number of metabolites above background in 10,000 pipetted WBM cell samples at various times after the death of the mouse (one experiment with n = 5 replicates per time point). (M) Number of metabolites that significantly increased or decreased at each time point relative to the 5 min time point (the metabolites are listed in *Figure 1—source data 6*, Supplementary Table 5). (N–P) Log2-transformed (N), non-transformed (O), and non-transformed intensity values for metabolites $<1\times10^8$ (P) in the 5 versus 120 min samples. The statistical significance of differences between treatments was assessed using a paired t-test (A), a Kolmogorov–Smirnov test (B) followed by Holm–Sidak's multiple comparisons adjustment, repeated measures one-way ANOVA followed by Tukey's (D) or Dunnett's (G) multiple comparisons adjustment, repeated measures two-way ANOVA followed by Sidak's multiple comparisons adjustment (F), or Spearman correlation analysis (J, K, N–P). All statistical tests were two-sided. Data represent mean ± SD. See also *Figure 1—figure supplement 1*.

The online version of this article includes the following source data and figure supplement(s) for figure 1:

**Source data 1.** All source data for *Figure 1*.
**Source data 2.** Supplementary table 1A-B.
**Source data 3.** Supplementary table 2A-B.
**Source data 4.** Supplementary table 3A-B.
**Source data 5.** Supplementary table 4A-B.
**Source data 6.** Supplementary table 5.
**Source data 7.** Supplementary table 6A-B.
**Source data 8.** The raw metabolomic analyses from experiments comparing AML and ALL cells (*Figure 1H–K*).
**Figure supplement 1.** Chromatographic performance of lipids separated by HILIC.
**Figure supplement 1—source data 1.** Source data for *Figure 1—figure supplement 1*.

concentrator, with drying in a new vacuum concentrator in a HEPA-filtered PCR hood, or without drying. We detected 98, 123, or 130 metabolites significantly above sheath fluid background (always fold change >2 and false discovery rate [FDR] < 0.05) in the samples dried in a standard vacuum concentrator, a HEPA-filtered vacuum concentrator, or undried, respectively (*Figure 1C*). We thus detected more metabolites above background in low abundance samples if we avoided sample drying and incorporated this approach into the method.

## Acetonitrile extraction

Metabolites are most commonly extracted from cells using miscible aqueous-organic solvents, with the elimination of proteins, non-soluble components, and cellular debris by centrifugation. Different metabolites require different solvents for extraction (*Rabinowitz and Kimball, 2007*). To test different solvents, we extracted metabolites from 100,000 pipetted WBM cells using 80% methanol in water, 40:40:20 acetonitrile:methanol:water, or 80% acetonitrile in water. Using HILIC and orbitrap mass spectrometry we detected an average of 317 metabolites in samples extracted with 80% acetonitrile and 266 or 273 metabolites in samples extracted with 80% methanol or 40:40:20 methanol:acetonitrile:water, respectively (*Figure 1D*). While we observed considerable overlap in the metabolites detected using each solvent, 80% acetonitrile yielded a number of metabolites that were not detected using the other solvents (*Figure 1E*). We thus selected 80% acetonitrile for further method development.

## Ion suppression and cell numbers

Ion suppression of metabolite signals can occur as a result of the salt in the phosphate buffered saline (PBS) sheath fluid used for flow cytometric sorting: 1–3 nl of sheath fluid is sorted along with each cell depending on whether a 70 μm or 100 μm nozzle is used. Flow cytometry sheath fluid must contain salt to electrostatically charge droplets for sorting; sorting more cells also sorts more salt. When using reverse phase chromatography, we reduced ion suppression by using 0.5× PBS sheath fluid and a 70 μm nozzle in four-way purity sort mode to minimize droplet volume (*Agathocleous et al., 2017*). After changing to HILIC, we retested whether 0.5× PBS or the 70 μm nozzle affected the number of metabolites we detected from 10,000 WBM cells. Sorting with the 70 μm nozzle increased the number of metabolites we could detect above sheath fluid background as compared to the 100 μm nozzle, regardless of PBS concentration (*Figure 1F*). We found no significant difference in the number of metabolites detected above background using 0.5× (157 ± 2) versus 1.0× (149 ± 9) PBS (*Figure 1F*).

We found 18 metabolites that significantly differed among samples sorted with 0.5× versus 1× PBS sheath fluid (always fold change >2 and FDR < 0.05; see the list of metabolites in *Figure 1— source data 2*, Supplementary Table 1). Pathway enrichment analysis did not identify any pathways that were significantly enriched among the changed metabolites. It is unlikely that these differences are a consequence of hypotonic shock because the laminar flow within the cytometer minimizes the mixing of the cell sample buffer (1× Hank's Buffered Salt Solution [HBSS]) and the sheath fluid. Cells pass through the flow cytometer in less than a second and are immediately lysed upon sorting into the extraction solvent. Differences between 0.5× and 1× PBS sheath fluid are more likely to reflect reduced ion suppression in samples sorted with 0.5× PBS and altered metabolite extraction efficiency. We selected 0.5× PBS as the sheath fluid for further method development as we tended to detect somewhat more metabolites and higher levels of some metabolites when using 0.5× PBS.

Next we tested if the number of metabolites we detected above background increased with increasing numbers of cells. We pipetted 10,000, 20,000, 30,000, 50,000, or 100,000 WBM cells (in equal volumes of HBSS buffer) directly into 80% acetonitrile and quantitated metabolites. The number of metabolites detected above sheath fluid background increased significantly with increasing numbers of cells, from 157 ± 4 metabolites in 10,000 cells to 222 ± 9 metabolites in 100,000 cells (*Figure 1G*). In the same experiment, we detected an average of 155 ± 2 metabolites from 10,000 flow cytometrically sorted WBM cells (*Figure 1G*). We thus detected similar numbers of metabolites in flow cytometrically sorted and unsorted samples.

## Effect of flow cytometry on metabolite levels

To determine if metabolic differences between cells are preserved during cell sorting using the methods described above, we sorted or pipetted 10,000 HNT-34 AML cells or 10,000 DND-41 T-ALL cells into 80% acetonitrile. We detected 143–167 metabolites above background in each sample. Principal component analysis revealed differences between sorted and pipetted AML cells (*Figure 1H*; see the list of metabolites in *Figure 1—source data 3*, Supplementary Table 2) whereas differences among sorted and pipetted ALL cells were subtle (*Figure 1H*; see the list of metabolites in *Figure 1—source data 4*, Supplementary Table 3). Forty-three metabolites significantly differed between sorted and pipetted AML cells while 19 metabolites differed between sorted and pipetted ALL cells (fold change >2 and FDR < 0.05). Pathway enrichment analysis revealed that metabolites that differed between sorted and pipetted AML cells were significantly enriched in 'cysteine and methionine metabolism'. No pathways were significantly enriched among metabolites that differed between sorted and pipetted ALL cells.

Irrespective of whether cells were sorted or pipetted, similar differences were observed between AML and ALL cells. Among sorted samples, 85 metabolites significantly differed between AML and ALL cells while among pipetted samples, 71 of the same metabolites differed (*Figure 1I*; see the list of metabolites in *Figure 1—source data 5*, Supplementary Table 4). Approximately 84% of the metabolites that significantly differed among sorted cells also significantly differed among pipetted cells and 73% of the significant differences among pipetted samples also significantly differed among sorted samples. Of the 14 metabolites that significantly changed in sorted but not pipetted cells, 12 trended in the same direction in both sets of samples. Of the 26 metabolites that changed in pipetted but not sorted samples, 19 trended in the same direction in both sets of samples. Thus,

most metabolites exhibited similar differences among AML and ALL cells irrespective of whether the cells were sorted.

To more systematically assess the similarity of pipetted and sorted samples, we plotted log2-transformed fold change values between AML and ALL cells for all metabolites above background in sorted versus pipetted samples (*Figure 1J*). The slope of the regression was near 1 (y = 0.96 × – 0.06) and the correlation was strong for most metabolites (Spearman correlation coefficient, r = 0.81). When we restricted the analysis to metabolites that significantly differed between sorted AML and ALL cells (fold change >2, FDR < 0.05), the correlation was even stronger (y = 0.95 × – 0.15; r = 0.92; *Figure 1K*). While the levels of some metabolites did change during sorting, most metabolites strongly correlated in sorted and unsorted samples.

## Effect of time on metabolite levels

It typically took up to 2 hr to sort HSCs into acetonitrile, starting from when the mice were killed. We wondered to what extent metabolite levels changed over time during cell isolation. To test this, we quickly flushed bone marrow from the long bones and made single cell suspensions in HBSS that we kept on ice. We pipetted 10,000 cell aliquots of WBM cells into acetonitrile at 5, 15, 30, 60, 120, 180, and 240 min after killing the mice then performed metabolomic analysis on each sample. At all time points, we detected 170–179 metabolites above sheath fluid background (*Figure 1L*). Relative to the samples collected at 5 min, only two metabolites significantly changed (fold change >2 and FDR < 0.05) in the samples collected at 15 min (*Figure 1M*). The number of metabolites that significantly changed increased over time, but most of the changes occurred by 120 min (*Figure 1M*; see the list of metabolites that changed over time in *Figure 1—source data 6*, Supplementary Table 5). The only metabolic pathway that was significantly enriched among metabolites that changed over time was 'purine metabolism' as purines were enriched in samples that incubated longer on ice. Thus, some metabolites did change over time but these represented less than 20% of detected metabolites.

To more broadly assess the similarity of the samples over time, we plotted log2-transformed values for all detected metabolites in 5 min versus 120 min samples (*Figure 1N*). The slope of the regression was near 1 (y = 0.97 × +0.62) and the correlation was strong, r = 0.98. We also plotted non-transformed values for all detected metabolites in 5 min versus 120 min samples, observing a similarly high correlation (*Figure 1O*). Finally, to most clearly show the differences between 5 and 120 min samples, we plotted only metabolites with signal intensity $<1 \times 10^8$ (*Figure 1P*). Again, the slope of the regression was near 1 (y = 0.95 × +1.1 × 10⁶) and the correlation was strong, r = 0.97. Thus, most metabolite intensity values strongly correlated among samples that incubated on ice for different periods of time. However, some metabolic changes may occur within seconds of harvesting a tissue (*Lu et al., 2017*) in a way that was not reflected in this experiment as the 5 min time point was the earliest at which a bone marrow cell suspension could be reliably obtained.

## Effect of cell suspension buffer on metabolite levels

We typically prepare hematopoietic cell suspensions using HBSS, which contains glucose. To test if this affected metabolite levels, we prepared bone marrow cell suspensions in HBSS or PBS then sorted 10,000 cells for metabolomic analysis. We detected 150–162 metabolites above background in these samples but only five significantly differed between cells isolated from HBSS versus PBS suspended samples (*Figure 1—source data 7*, Supplementary Table 6). The metabolite that most differed between these samples was glucose, which was substantially enriched in cells isolated from HBSS. Pathway enrichment analysis did not detect any pathways enriched among the metabolites that differed among HBSS and PBS samples. Therefore, the presence of glucose in the cell suspension buffer did affect glucose levels in the sorted cells but had little effect on the levels of other metabolites.

## Metabolomic profiling of HSC/multipotent progenitors (MPPs)

To assess the metabolite profile of HSCs/MPPs we sorted 10,000 cell aliquots of CD48⁻Lineage⁻Sca1⁺c-kit⁺ cells and WBM cells. CD48⁻Lineage⁻Sca1⁺c-kit⁺ cells represent 0.05% of WBM cells and are a very highly enriched for HSCs and MPPs (*Oguro et al., 2013*). The metabolite profiles of HSCs and MPPs are extremely similar (*Agathocleous et al., 2017*). We detected 160 ± 15

metabolites above sheath fluid background in HSCs/MPPs and 147 ± 15 in WBM (*Figure 2A*). A total of 78 metabolites significantly differed in abundance between HSCs/MPPs and WBM cells (fold change >2 and FDR < 0.05; the metabolites are listed in *Figure 2—source data 2*, Supplementary Table 1). Of these, 51 differed by at least 2.5-fold (*Figure 2B*). Of the 16 metabolites that *Agathocleous et al., 2017* found to significantly differ between HSCs/MPPs and WBM cells, 13 also significantly differed, in the same direction, using the new method (*Figure 2—figure supplement 1*). The other three metabolites either were not detected using the new method or could not be quantitated accurately due to extraction conditions. Thus, the new method detected most of the metabolic differences between HSC/MPPs and WBM cells observed by *Agathocleous et al., 2017*, while also detecting 65 additional differences.

Pathway enrichment analysis found only one pathway that was significantly enriched (FDR < 0.01): 10 of 36 metabolites in the murine KEGG 'glycerophospholipid metabolism' pathway significantly differed in abundance between HSC/MPPs and WBM cells. HSCs were enriched for many components of the Kennedy (cytidine diphosphate-choline) pathway (*Li and Vance, 2008*; *Kennedy and Weiss, 1956*), including choline, choline phosphate, CDP-choline, ethanolamine phosphate, glycerophosphorylcholine, glycerophosphorylethanolamine, and many phosphatidylcholines (PC),

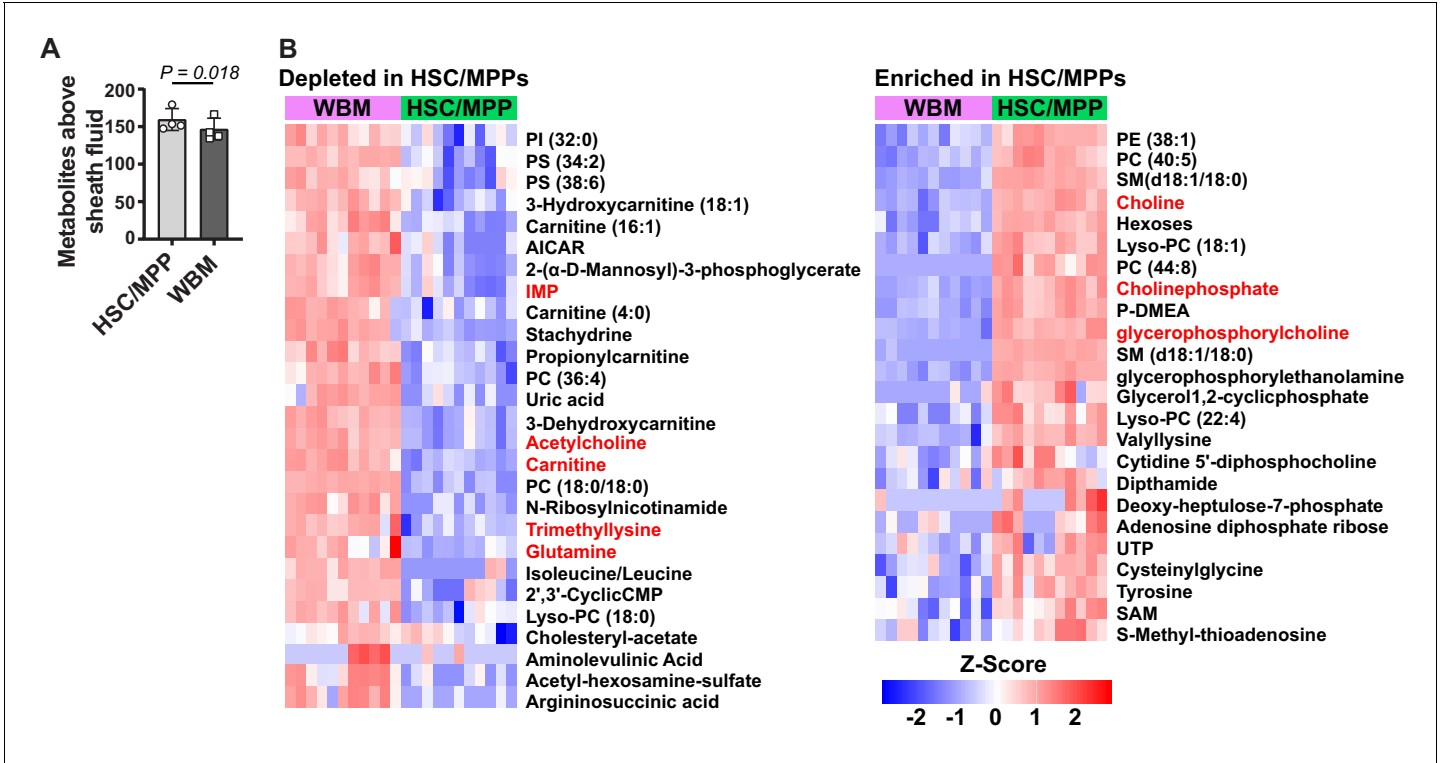

**Figure 2.** Metabolic differences between hematopoietic stem cell (HSC)/multipotent progenitors (MPPs) and whole bone marrow (WBM) cells. (**A**) Metabolites significantly above background in 10,000 sorted HSC/MPPs or WBM cells (n = 3–7 replicates per treatment in each of four independent experiments; fold change >2 and false discovery rate [FDR] < 0.05). (**B**) Metabolites that were significantly depleted (left) or enriched (right) in HSC/MPPs as compared to WBM cells (fold change >2.5, FDR < 0.01; all metabolites with fold change >2.0 and FDR < 0.05 are listed in *Figure 2—source data 2*, Supplementary Table 1). Data in (**A**) represent mean ± SD. A comparison of these differences to those observed by *Agathocleous et al., 2017* between HSCs and WBM cells is shown in *Figure 2—figure supplement 1* and a summary of the differences in lipid species is shown in *Figure 2— figure supplement 2*.

The online version of this article includes the following source data and figure supplement(s) for figure 2:

**Source data 1.** All source data for *Figure 2*.

**Source data 2.** Supplementary table 1.

**Figure supplement 1.** Metabolites that were detected as differing between hematopoietic stem cells (HSCs)/multipotent progenitors (MPPs) and whole bone marrow (WBM) cells using the Agathocleous et al.'s method (*Agathocleous et al., 2017*) versus the method described in this study.

**Figure supplement 2.** Glycerophospholipids are enriched in hematopoietic stem cell (HSC)/multipotent progenitors (MPPs) as compared to whole bone marrow (WBM) cells.

phosphatidylethanolamines (PE), lysophosphatidylcholines (Lyso-PC), and lysophosphatidylethanol-amines (Lyso-PE) (*Figure 2—figure supplement 2B*). Acetylcholine and several phosphatidylserine (PS) species were depleted in HSC/MPPs as compared to WBM (*Figure 2—figure supplement 2B*). These results raise the possibility that glycerophospholipid synthesis is activated in HSC/MPPs relative to WBM; however, additional studies will be required in the future to test this. The prominence of phospholipids among the differences between HSCs/MPPs and WBM cells illustrates the ability of the new method to detect differences not detected by prior methods.

To determine whether metabolic perturbations in HSCs in vivo can be detected by this method, we treated mice for 3 days with methotrexate. Methotrexate inhibits dihydrofolate reductase (DHFR) and AICAR transaminase (ATIC), steps in de novo purine biosynthesis (*Baggott et al., 1986*). Methotrexate treatment did not significantly affect bone marrow cellularity or the frequencies of HSCs, MPPs, or LSK cells in the bone marrow (*Figure 3A–D*). Methotrexate treatment also did not significantly affect the reconstituting potential of WBM cells upon competitive transplantation into irradiated mice (*Figure 3E*). Metabolomic analysis of 10,000 HSC/MPPs from the bone marrow of methotrexate-treated and control mice revealed that the only pathway that was significantly enriched among the metabolites that differed was 'purine metabolism'. While methotrexate would

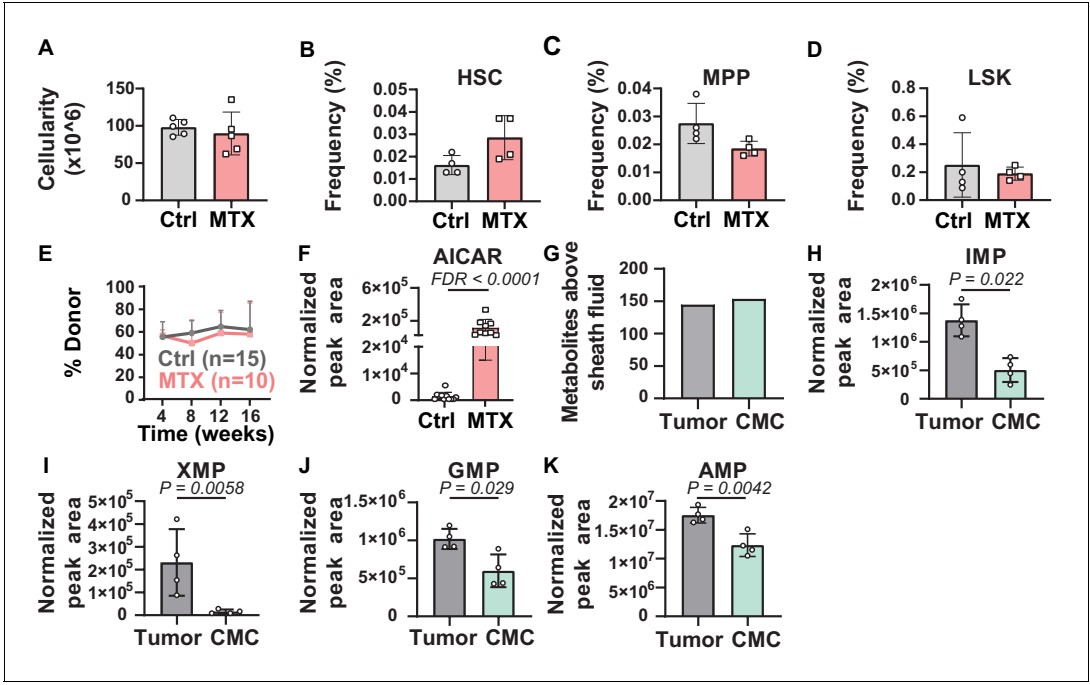

**Figure 3.** Metabolic differences between methotrexate-treated and control hematopoietic stem cells (HSCs) or circulating melanoma cells and primary tumors. (A–D) Bone marrow cellularity (A) and the frequencies of CD150$^+$CD48$^-$Lin$^-$Sca1$^+$c-kit$^+$ HSCs (B), CD150$^-$CD48$^-$Lin$^-$Sca1$^+$c-kit$^+$multipotent progenitors (MPPs) (C), and Lin$^-$Sca1$^+$c-kit$^+$ cells (D) in femurs and tibias from mice treated with methotrexate or vehicle control (n = 5 mice per treatment from two independent experiments). (E) Percentage of nucleated blood cells that were donor-derived after competitive transplantation of bone marrow cells from methotrexate-treated versus control mice into irradiated recipients (two independent experiments). (F) AICAR levels in HSC/MPPs from mice treated with methotrexate or vehicle (11 control samples and 9 MTX samples from four independent experiments). (G) Metabolites detected above background in primary tumor cells or circulating melanoma cells (n = 3 or four replicates per treatment in one experiment; fold change >2 and false discovery rate [FDR] < 0.05). (H–K) Levels of the purines inosine monophosphate (IMP) (H), xanthosine monophosphate (XMP) (I), guanosine monophosphate (GMP) (J), and adenosine monophosphate (AMP) (K) in primary tumor and circulating melanoma cells. Statistical significance was assessed by t-test (A), repeated measures two-way ANOVA (B–D), or mixed effects analysis (E) followed by Sidak's multiple comparisons adjustment. All tests were two-sided. Data represent mean ± SD. The flow cytometry gates used to isolate each cell population are shown in *Figure 3—figure supplement 1*. All of the metabolites that differed between circulating melanoma cells and subcutaneous tumor cells are listed in *Figure 3—source data 2*, Supplementary table 1.

The online version of this article includes the following source data and figure supplement(s) for figure 3:

**Source data 1.** All source data for *Figure 3*.
**Source data 2.** Supplementary table 1.
**Figure supplement 1.** Flow cytometry gating strategies.

also be expected to alter folate metabolism, folate species are very difficult to detect by metabolomics (*Zheng et al., 2018*; *Chen et al., 2017*) and are not detected by our method. Given that methotrexate inhibits ATIC, AICAR levels would be expected to increase after methotrexate treatment (*Cronstein et al., 1993*; *Baggott et al., 1986*; *Allegra et al., 1985*). Consistent with this, AICAR levels were 88-fold higher in HSCs/MPPs from methotrexate-treated as compared to control mice (*Figure 3F*). The method was thus capable of detecting expected metabolic perturbations in HSCs in vivo.

## Metabolomic profiling of circulating cancer cells

To test if the method is broadly applicable, we tested if we could detect metabolic differences between circulating melanoma cells from the blood and the primary subcutaneous tumors from which they arose. When efficiently metastasizing human melanomas are subcutaneously transplanted into NSG mice they spontaneously metastasize, giving rise to rare circulating melanoma cells in the blood, lymph, and metastatic tumors (*Piskounova et al., 2015*; *Tasdogan et al., 2020*; *Ubellacker et al., 2020*). We subcutaneously transplanted M405 patient-derived melanoma cells into NSG mice. When the subcutaneous tumors reached 2.5 cm in diameter, we isolated 10,000 cell aliquots of melanoma cells by flow cytometry from mechanically dissociated subcutaneous tumors as well as from the blood of the same mice. We pooled blood from 6 to 10 mice per sample to isolate 10,000 circulating melanoma cells.

We detected 145 and 154 metabolites above sheath fluid background in the subcutaneous tumor and circulating melanoma cell samples, respectively (*Figure 3G*). Pathway enrichment analysis of all metabolites that significantly differed between subcutaneous tumor and circulating melanoma cells ($p < 0.05$) found one pathway that significantly (FDR < 0.01) differed, 'purine metabolism'. Several purine biosynthesis intermediates were depleted in circulating melanoma cells as compared to subcutaneous tumors, including insoine monophosphate (IMP), xanthosine monophosphate (XMP), guanosine monophosphate (GMP), and adenosine monophosphate (AMP) (*Figure 3H–K*; *Figure 3— source data 2*, Supplementary table 1). Given that circulating melanoma cells experience high levels of oxidative stress (*Piskounova et al., 2015*; *Tasdogan et al., 2020*), these data raise the possibility that metastasizing melanoma cells reduce purine biosynthesis, and perhaps other anabolic pathways, to preserve NADPH for oxidative stress resistance.

## Discussion

The method for metabolomic analysis of rare cells described in this study significantly increased metabolite numbers and pathway coverage relative to our prior method (*Agathocleous et al., 2017*; *Figure 4A and B*). We improved signal to noise ratio by using HILIC and an orbitrap mass spectrometer. We decreased contamination by eliminating sample drying and improved chromatographic performance by extracting metabolites with 80% acetonitrile. In principle, this method can be used to analyze any cell population isolated by flow cytometry, though in practice it is most useful when cell numbers are limited and when enzymatic dissociation is not required. We normalized for input variation among samples by ensuring that the average signal intensity values of metabolites detected above background were equal in the samples being compared. While it is impractical to include isotopically labeled internal standards within the initial metabolomic analysis, since the relative levels of hundreds of different metabolites are assessed, we routinely follow-up metabolomics with other methods optimized to extract and quantitate specific metabolites of interest. In those follow-up assays we include labeled internal standards to determine absolute concentrations (*Tasdogan et al., 2020*; *Ubellacker et al., 2020*; *Piskounova et al., 2015*).

Cells can undergo metabolic changes upon removal from their in vivo environment (*Lau et al., 2020*). This is a particular problem when cells are enzymatically dissociated as they exchange metabolites with the dissociation medium, or when cells are sorted into buffers that require additional processing steps before cell lysis and metabolite extraction (*Lau et al., 2020*; *Binek et al., 2019*; *Llufrio et al., 2018*). To avoid changes in metabolites during cell processing, we worked quickly and kept the cells cold from the time they left the animal until they were sorted into acetonitrile. Cellular metabolism is immediately quenched by sorting into cold acetonitrile. The levels of most metabolites strongly correlated in sorted and unsorted samples (*Figure 1H–K*). Some metabolites exhibited changes over time during cell processing (*Figure 1M*) but the levels of most metabolites strongly

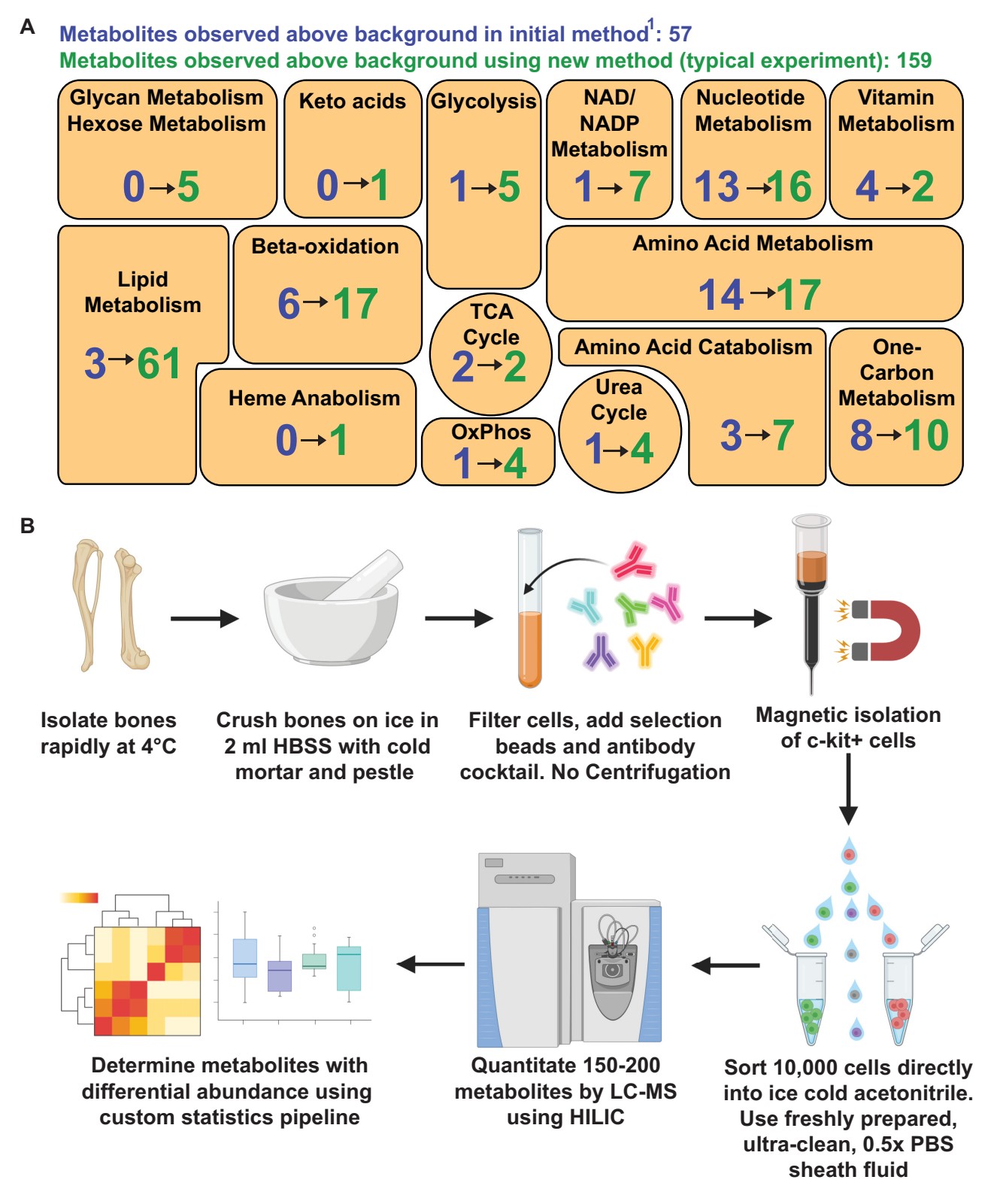

**Figure 4.** Metabolomic profiling of hematopoietic stem cells (HSCs) isolated by flow cytometry. (A) Overview of the method. (B) Metabolites detected above background in 10,000 HSCs/multipotent progenitors (MPPs) in this study (green numbers, 159 metabolites total) as compared to our prior study using a different method (*Agathocleous et al., 2017*) (blue numbers, 57 metabolites total). These data are from one experiment, representative of four

*Figure 4 continued on next page*

*Figure 4 continued*

independent experiments. Metabolites detected above background were calculated by comparing three whole bone marrow (WBM) or three HSC/MPP samples to three sheath fluid blanks (fold change >2, false discovery rate [FDR] < 0.05).

The online version of this article includes the following source data for figure 4:

**Source data 1.** All source data for *Figure 4*.

correlated in samples at 5 and 120 min after removal from the mouse (*Figure 1N–P*). Therefore, most metabolite levels were not significantly changed by cell preparation and sorting, at least beyond 5 min after the mouse was killed. However, some metabolite levels might change less than a minute after the removal of cells from their normal physiological environment in a way that would not have been detected in our experiments (*Lu et al., 2017*). If so, it will not be possible to quantitate these metabolites in flow cytometrically isolated cells.

Transitioning to HILIC provided several advantages relative to reverse phase chromatography. We were able to avoid sample drying, which significantly reduced contamination (*Figure 1B and C*). We reduced ion suppression compared to our prior method because salt eluted after the metabolites of interest on HILIC columns (*Figure 1F*). Third, HILIC improved the separation of polar metabolites, such as central carbon metabolites, while also enabling the detection of many lipid species. Nonetheless, some metabolites are better resolved and detected by reverse phase chromatography. Therefore, while HILIC provided a net advantage for our purposes, adapting this low cell number method to other chromatographies could improve the detection of certain classes of metabolites.

Prior studies have explored many aspects of metabolomics methods, including extraction solvents and drying (*Theodoridis et al., 2012*). Methanol–water and acetonitrile–water mixtures have been reported to capture more metabolites than other solvents (*Want et al., 2006*; *Jiye et al., 2005*; *Bruce et al., 2008*; *Masson et al., 2010*). Methanol is nucleophilic and can degrade metabolites with electrophilic moieties, such as nucleoside phosphates (*Rabinowitz and Kimball, 2007*). Data quality and accuracy are also improved by minimizing sample manipulation after metabolite extraction, including by avoiding drying, which can promote oxidation (*Siegel et al., 2014*; *Fan et al., 2014*; *Chen et al., 2017*; *Lu et al., 2017*). We found that acetonitrile–water solvent and avoiding drying yielded the highest number of metabolites detected above sheath fluid background. HILIC resolution of metabolites was also improved when samples were injected in acetonitrile–water instead of methanol–water.

The extraction conditions we used are not suitable for the quantitation of some metabolites, including those that spontaneously oxidize (*Lu et al., 2018*). For example, ascorbate spontaneously oxidizes upon extraction from cells *Washko et al., 1992*; therefore, in our prior study we added EDTA to the extraction solvent to prevent spontaneous oxidation (*Agathocleous et al., 2017*). In an effort to devise a general method in the current study, we did not add EDTA to the extraction solvent and therefore did not measure ascorbate levels accurately. Consistent with this, ascorbate was one of the three metabolites that differed between HSCs and WBM cells in our prior study (*Agathocleous et al., 2017*) that we did not detect as different in the current study (*Figure 2—figure supplement 1*). The other two were spermidine and betaine, which were not detected using the new method.

We observed differences in the abundance of glycerophospholipids between HSCs/MPPs and WBM cells. Functional studies will be required to assess the biological significance of this difference. Few studies have examined lipid metabolism in HSCs (*Xie et al., 2019*; *Ito et al., 2012*; *Ito et al., 2016*; *Lee et al., 2018*; *Pernes et al., 2019*), partly because methods have not been readily available to quantitate lipid levels in HSCs. The ability of the new method to detect more than 60 lipids in 10,000 HSCs may facilitate future studies of lipid metabolism in stem cells. We also performed metabolomics on circulating melanoma cells from xenografted mice. Cancer cells must undergo metabolic changes to survive oxidative stress during metastasis (*Piskounova et al., 2015*; *Tasdogan et al., 2020*). Better understanding the metabolic changes could reveal new therapeutic vulnerabilities to block cancer progression.

# Materials and methods

## Key resources table

| Reagent type (species) or resource | Designation | Source or reference | Identifiers | Additional information |
|---|---|---|---|---|
| Chemical compound, drug | Phosphate Buffered Saline Tablet | Sigma | Cat# P4417-100TAB | |
| Chemical compound, drug | Methanol, Optima grade for LC/MS | Fisher Scientific | Cat# A456-4 | |
| Chemical compound, drug | Acetonitrile, Optima Grade for LCMS | Fisher Scientific | Cat# A955-4 | |
| Chemical compound, drug | MeOH, Optima Grade for LCMS | Fisher Scientific | Cat# A456-4 | |
| Chemical compound, drug | Water, Optima Grade for LCMS | Fisher Scientific | Cat# W6-4 | |
| Commercial assay, kit | LS magnetic enrichment columns | Miltenyi | Cat# 130-042-401 | |
| Commercial assay, kit | MidiMACS separator | Miltenyi | Cat# 130-042-301 | |
| Other | Plastic microfuge tube opener | USA Scientific | Cat# 1400–1508 | For minimizing metabolite contamination when opening microfuge tubes |
| Antibody | FITC anti-mouse B220, clone: RA3-6B2 (rat monoclonal) | Tonbo | Cat# 35–0452 U500, RRID:AB_2621690 | Dilution: 1:400 For bone marrow HSC isolation by flow cytometry |
| Antibody | FITC anti-mouse Gr-1, clone: RB6-8C5 (rat monoclonal) | Tonbo | Cat# 35–5931 U500, RRID:AB_2621721 | Dilution: 1:400 For bone marrow HSC isolation by flow cytometry |
| Antibody | FITC anti-mouse Terr119, clone: TER-119 (rat monoclonal) | Tonbo | Cat# 35–5921 U500, RRID:AB_2621720 | Dilution: 1:400 For bone marrow HSC isolation by flow cytometry |
| Antibody | FITC anti-mouse CD2, clone: RM2-5 (rat monoclonal) | Tonbo | Cat# 35–0021 T100, RRID:AB_2621657 | Dilution: 1:400 For bone marrow HSC isolation by flow cytometry |
| Antibody | FITC anti-mouse CD3, clone: 17A2 (rat monoclonal) | Tonbo | Cat# 35–0032 U100, RRID:AB_2621660 | Dilution: 1:400 For bone marrow HSC isolation by flow cytometry |
| Antibody | FITC anti-mouse CD5, clone: 53–7.3 (rat monoclonal) | Biolegend | Cat# 100606, RRID:AB_312735 | Dilution: 1:400 For bone marrow HSC isolation by flow cytometry |
| Antibody | FITC anti-mouse CD8$\alpha$, clone: 53–6.7 (rat monoclonal) | Tonbo | Cat# 35–0081 U500, RRID:AB_2621671 | Dilution 1:400 For bone marrow HSC isolation by flow cytometry |
| Antibody | APC-e780 anti-mouse c-kit, clone: 2B8 (rat monoclonal) | eBiosciences | Cat# 47-1171-82, RRID:AB_1272177 | Dilution: 1:200 For bone marrow HSC isolation by flow cytometry |
| Antibody | PerCP-Cy5.5 anti-mouse Sca-1, clone: D7 (rat monoclonal) | BioLegend | Cat# 108124, RRID:AB_893615 | Dilution 1:200 |
| Antibody | APC anti-mouse CD48, clone: HM48-1 (Armenian hamster monoclonal) | eBiosciences | Cat# 17-0481-82, RRID:AB_469408 | Dilution 1:200 |
| Antibody | PE anti-mouse CD150, clone: TC15-12F12.2 (rat monoclonal) | BioLegend | Cat# 115904, RRID:AB_313683 | Dilution 1:200 |
| Antibody | APC anti-Mouse CD45, clone: 30-F11 (rat monoclonal) | Tonbo | Cat# 20–0451 U100, RRID:AB_2621573 | Dilution 1:100 |

*Continued on next page*

*Continued*

| Reagent type (species) or resource | Designation | Source or reference | Identifiers | Additional information |
|---|---|---|---|---|
| Antibody | APC anti-Mouse CD31 (PECAM-1), clone: 390 (rat monoclonal) | Biolegend | Cat# 102410, RRID:AB_312905 | Dilution 1:100 |
| Antibody | APC anti-Mouse Ter119, clone: Ter119 (rat monoclonal) | Tonbo | Cat# 20–5921 U100, RRID:AB_2621609 | Dilution 1:100 |
| Antibody | FITC anti-Human HLA-A, B, C, clone: G46-2.6 (mouse monoclonal) | BD Biosciences | Cat# 555552, RRID:AB_395935 | Dilution 1:20 |
| Antibody | Anti-Mouse c-Kit, conjugated to para-magnetic beads, clone: 3C11 (rat monoclonal) | Miltenyi | Cat# 130-091-224, RRID:AB_2753213 | (60 µl) 60 µl beads per $3 \times 10^8$ whole bone marrow cells |
| Antibody | Anti-Mouse CD45, conjugated to para-magnetic beads, clone: 30-F11 (rat monoclonal) | Miltenyi | Cat# 130-052-301, RRID:AB_2877061 | (3 µl) 3 µl beads per $1 \times 10^6$ whole bone marrow cells |
| Chemical compound, drug | DAPI | Sigma-Aldrich | Cat# D8417-10mg | 1 µg/ml for flow cytometry |
| Cell line | Human – HNT-34 AML cell line | Provided by Jian Xu's Laboratory at Children's Research Institute at UT Southwestern Medical Center. Original source: DSMZ | ACC 600, RRID: CVCL_2071 | |
| Cell line | Human DND-41 cell line | Provided by Jian Xu's Laboratory at Children's Research Institute at UT Southwestern Medical Center. Original source: Hui Feng, M.D./Ph.D.; Boston University | RRID: CVCL_2022 | |
| Cell line | Human melanoma xenograft M405 | Sci Trans Med 4:159ra PMCID:PMC4501487 | M405 | |
| Strain, strain background | NOD.CB17-Prkdcscid Il2rgtm1Wjl/Szj (NSG) mice | Jackson laboratories | 005557, RRID:IMSR_JAX:005557 | |
| Strain, strain background | C57BL/Ka Thy1.1 mice | Derived from Henry Kaplan's laboratory at Stanford University | N/A | |
| Other | ZIC-pHILIC column (2.1 × 150, 5 µm) | Millipore Sigma | Cat# 1504600001 | |
| Software, algorithm | Omics Data Analyzer (ODA) | This manuscript | https://git.biohpc.swmed.edu/CRI/ODA | See Materials and methods, section 'Statistical analysis of metabolomics data' |
| Software, algorithm | Graphpad Prism V8.3 | Graphpad | RRID:SCR_002798 | |
| Software, algorithm | FlowJo V10.7.1 | BD Biosciences | RRID:SCR_008520 | |
| Software, algorithm | Freestyle V1.5 | Thermo Scientific | N/A | |
| Software, algorithm | Trace Finder V4.0 | Thermo Scientific | N/A | |
| Software, algorithm | Compound Discoverer V3.1 | Thermo Scientific | N/A | |
| Chemical compound, drug | Formic Acid Optima | Fisher | Cat# A11750 | |
| Chemical compound, drug | Ammonium acetate, Optima | Fisher | Cat# A11450 | |

*Continued on next page*

*Continued*

| Reagent type (species) or resource | Designation | Source or reference | Identifiers | Additional information |
|---|---|---|---|---|
| Chemical compound, drug | Ammonium hydroxide, Optima | Fisher scientific | Cat# A470-250 | |
| Chemical compound, drug | Methotrexate | Selleck Chem | Cat# S1210 | |
| Chemical compound, drug | SplashMix | Avanti | Cat# 330707 | |

## Isolation of hematopoietic cells for metabolomics

Bone marrow cells were collected quickly and maintained at 0–4°C throughout the staining and isolation procedure to minimize metabolic changes. Mice were euthanized by cervical dislocation. Bones were rapidly dissected and stored on ice in HBSS without $Mg^{2+}$ and $Ca^{2+}$ (HBSS, Corning). Muscle was stripped from the bones, then they were crushed in 2.5 ml of HBSS using a pre-cooled mortar and pestle, on ice. Bone marrow cells were filtered through a 40 µm strainer into a 50 ml conical tube. The cells were then stained with fluorochrome-conjugated antibodies against B220 (FITC, Tonbo), Gr-1 (FITC, Tonbo), Ter119 (FITC, Tonbo), CD2 (FITC, Tonbo), CD3 (FITC, Tonbo), CD5 (FITC, BioLegend), CD8 (FITC, Tonbo), c-kit (APC-eFluor780, eBiosciences), Sca-1 (PerCP-Cy5.5, BioLegend), CD48 (APC, eBiosciences), and CD150 (PE, BioLegend) for 30 min on ice. Beginning 10 min before adding the antibodies and continuing after the antibodies had been added, para-magnetic beads conjugated to anti-c-kit antibodies (Miltenyi) were added to the cells to facilitate pre-enrichment of c-kit$^+$ cells in samples from which HSCs were sorted. To ensure that WBM cells were processed in the same way, these samples were enriched by positive selection of para-magnetic beads bound to anti-CD45 antibodies (Miltenyi). Positive selection was performed in the cold room at 4°C using a QuadroMACS manual separator (Miltenyi) and LS Columns (Miltenyi). Cells were eluted from columns in 2 ml of HBSS, centrifuged for 5 min at 300 × g, and resuspended in HBSS with 4′,6-diamidino-2-phenylindole (DAPI, 1 µg/ml, Sigma) for flow cytometry. The gating strategy for the isolation of HSCs/MPPs is depicted in *Figure 3—figure supplement 1A*.

## Isolation of melanoma cells for metabolomics

Mice were transplanted subcutaneously with human melanoma cells and the cells were allowed to spontaneously metastasize until the subcutaneous tumors reached 2.5 cm. At this point, single cell suspensions were obtained by dissociating tumors mechanically with a scalpel on ice followed by gentle trituration. Cells were filtered through a 40 µm strainer to generate a single cell suspension. Blood was collected from mice by cardiac puncture with a syringe pretreated with citrate–dextrose solution (Santa Cruz).

Subcutaneous tumor and blood specimens were first incubated on ice for 10 min with ammonium-chloride-potassium (ACK) lysing buffer to eliminate red blood cells. The cells were washed with PBS and then stained with antibodies prior to flow cytometry. All antibody staining was performed for 20 min on ice, followed by washing with PBS and centrifuging at 200 × g for 5 min. Cells were stained with directly conjugated antibodies against mouse CD45 (APC, Tonbo Biosciences), mouse CD31 (APC, Biolegend), mouse Ter119 (APC, Tonbo Biosciences), and human HLA-A, -B, -C (G46-2.6-FITC, BD Biosciences). Human melanoma cells were isolated as cells that were positive for HLA and DsRed (melanoma cells were tagged with constitutive DsRed before subcutaneous transplantation), and negative for mouse endothelial (CD31) and hematopoietic markers (CD45 and Ter119). Cells were washed with PBS and resuspended in DAPI (1 µg/ml, Sigma) to eliminate dead cells from sorts and analyses. The flow cytometry gating strategies for the isolation of primary tumor cells and circulating melanoma cells are depicted in *Figure 3—figure supplement 1B and C*.

## Cell lines

Human HNT-34 (DSMZ Cat # ACC 600, RRID: CVCL_2071) and Human DND-41 (Hui Feng, M.D./Ph.D.; Boston University, RRID: CVCL_2022) cell lines were provided by Jian Xu's laboratory at Children's Research Institute at UT Southwestern. To confirm the identity of these cell lines, we performed qRT-PCR and western blot analyses of several leukemia signature genes (*Ng et al., 2016*). We also

performed RNA-seq and whole genome sequencing and compared the results with previous studies. All cell lines tested negative for mycoplasma contamination using the Lonza MycoAlert kit (Lonza # LT07-118). No cell lines used in this study were found in the database of commonly misidentified cell lines maintained by the International Cell Line Authentication Committee (ICLAC) or the National Center for Biotechnology Information (NCBI) BioSample.

## Sorting versus pipetting of cultured cells for metabolomics

HNT-34 AML cells and DND-41 T-ALL cells were cultured non-adherently in RPMI with 10% fetal bovine serum and 1% penicillin/streptomycin. Cells were maintained at a density of $5 \times 10^5$ cells/ml, and cultured at 37°C with 5% $CO_2$. AML or ALL cells were removed from the incubator and centrifuged at 4°C to pellet the cells, then resuspended in ice-cold PBS and centrifuged again, before being resuspended in ice cold PBS at a density of $1 \times 10^6$ cells/ml. Ten microliters of the AML or ALL cell suspension was then pipetted into 40 µl of 100% acetonitrile to create a final metabolite extract of 10,000 cells in 50 µl 80% acetonitrile (10 min total processing time). Ten thousand AML or ALL cells from the same cell suspensions were also sorted into 40 µl of 100% acetonitrile to create a final metabolite extract of 10,000 cells in 50 µl of 80% acetonitrile (30–60 min processing time). The cells were kept ice cold before and during sorting.

## Flow cytometer preparation

Flow cytometers were thoroughly cleaned before sorting low abundance samples to minimize background. All flow cytometry was performed using a FACSAria II or a FACSAria Fusion (BD Biosciences). The fluidics shutdown protocols were performed using 80% ethanol before each sort. A clean, metabolomics-dedicated FACSAria sheath tank was rinsed with ultrapure water several times to reduce contamination, before being filled with 4 l of 0.5× PBS made from tablets (Sigma) dissolved in ultrapure water. The metabolomics sheath tank was connected to the sorter using a dedicated 0.22 µm filter. The fluidics startup protocol was performed using freshly made 0.5× PBS sheath fluid. The sorter was configured to use a 70 µm nozzle but before the nozzle was inserted two cycles of clean flow cell protocols were performed with Windex. The sheath fluid was then run through the flow cytometer without a nozzle for 5 min to flush Windex and any remaining debris from the flow cell. At the same time, the 70 µm nozzle was sonicated for 5 min to remove contamination and debris, and the cleanliness of the nozzle was confirmed by microscopy. The sheath fluid stream was turned off and the sort chamber was cleaned with a lint-free wipe and cotton swabs. The nozzle was then inserted and the stream was turned on. The sample line was cleaned again by running a 5 ml sample tube of Windex for 5 min, followed by ultrapure water for 5 min. Four-way purity sort mode was used to minimize droplet size. The cell sample, the sorting chamber, and the collection tube adapter were all maintained at 4°C during sorting.

## Sorting cells for metabolomics

The Eppendorf tubes into which cells were sorted were loaded with 40 µl of 100% acetonitrile (Optima, Fisher Scientific) or methanol (Optima, Fisher Scientific) before sorting. We used a freshly opened bag of clean Eppendorf tubes (USA Scientific) and filtered pipette tips. The Eppendorf tubes were maintained at −20°C until just prior to sorting. Cell samples were filtered through a 40 µm strainer before sorting. The flow rate was minimized to reduce shear stress. Just before sorting, the Eppendorf tubes were opened using a clean microfuge tube opener (USA Scientific) to avoid contamination. After sorting, the tubes were sealed, vortexed, and centrifuged briefly to collect all the liquid in the bottom of the tube, and placed on dry ice. Metabolites were extracted by vortexing again for 1 min at high speed, followed by centrifugation at 17,000 × g for 15 min at 4°C. The supernatant was transferred to auto-sampler vials with low volume inserts and analyzed immediately by LC-MS (see details below).

## Liquid chromatography and mass spectrometry

Liquid chromatography was performed with a Vanquish Flex UHPLC (Thermo Scientific). The reverse phase method used a Waters HSS C18 column (2.1 × 150 mm, 1.7 µm) with a binary solvent gradient. Mobile phase A was water with 0.1% formic acid and mobile phase B was acetonitrile with 0.1% formic acid. Gradient separation proceeded as follows: from 0 to 5 min, 0% B; from 5 min to 45 min

mobile phase B was ramped linearly from 0% to 100%; from 45 min to 52 min, mobile phase B was held at 100%; from 52 to 52.1 min, mobile phase B was ramped linearly to 0%; from 52.1 to 60 min, mobile phase B was held at 0%. Throughout the course of the method, the solvent flow rate was kept to 100 µl/min and column temperature was held at 30℃.

The HILIC method used a Millipore Sigma ZIC-pHILIC column (2.1 × 150, 5 µm) with a binary solvent gradient. Mobile phase A was water containing 10 mM ammonium acetate, pH 9.8 with ammonium hydroxide; mobile phase B was 100% acetonitrile. Gradient separation proceeded as follows: from 0 to 15 min mobile phase B was ramped linearly from 90% to 30%; from 15 min to 18 min, mobile phase B was held at 30%; from 18 min to 19 min, mobile phase B was ramped linearly from 30% to 90%; mobile phase B was held at 90% from 19 min to 27 min to regenerate the initial chromatographic environment. Throughout the method, solvent flow rate was kept at 250 µl/min and the column temperature was maintained at 25℃. For low abundance samples, 20 µl of sample was injected onto the column. For high abundance samples, 10 µl was injected.

All mass spectrometry data were acquired using a Thermo Scientific (Bremen, Germany) QExactive HF-X mass spectrometer (LC-MS/MS). For low abundance samples, polarity-switching MS1 only acquisition was used. Each polarity was acquired at a resolving power of 120,000 full width at half maximum (FWHM); the automatic gain control (AGC) target was set to 1,000,000 with a maximum inject time of 50 ms. The scan range was set to 80–1200 Daltons. High-abundance samples analyzed for library construction were acquired with two separate ddMS2 methods – one for positive mode and another for negative mode. Precursor MS1 data for this method were acquired with the exact same settings as those described above. Product ion MS data were acquired with a resolving power of 15,000 FWHM; the AGC target was set to 200,000, with a maximum inject time of 150 ms. A top-10 data dependent MS scheme was used with an isolation window of 1 Da and an isolation offset of 0.5 Da. Analytes were fragmented with stepped collision energies of 30, 50, and 70 normalized collision energy (NCE) units. The minimum AGC target was 8000 with a dynamic exclusion of 30 s.

Instrument performance was evaluated before each experiment by analyzing a quality control sample, 20 µl of freshly obtained rat serum. We compared peak areas of individual metabolites and the total number of metabolites detected in the control sample with control samples run in prior experiments. If the peak area values and total metabolite identifications fell outside of 1 standard deviation from the historical average, the instrument was cleaned to re-optimize sensitivity.

## Metabolite library development

To develop the metabolite library we used to analyze samples, we acquired LC-MS/MS data from high abundance samples using a data dependent MS/MS method. Metabolites were identified in an unbiased fashion using Compound Discoverer 3.0 (ThermoScientific). Metabolites were added to the initial library only if they met the following criteria. First, chromatographic peaks had to align in all samples, and peak intensity had to increase with cell number. Second, precursor mass accuracy had to be within 5 ppm of theoretical mass, with an naturally occurring isotope pattern that matched that predicted by the chemical formula. Third, the MS/MS product ion spectra had to either match an annotated database (mzCloud, Human Metabolome Data Base, Lipid Maps, and ChemSpider) or had to be confirmed by analysis of chemical standards. This process yielded a 590 metabolite library with known masses and chromatographic retention times. This library was imported into the manual peak review software Trace Finder 4.1 (ThermoScientific) for manual peak integration of all low abundance LC-MS data. To narrow this list of 590 metabolites to the metabolites that might be detected in 10,000 sorted cells, we determined which of the 590 metabolites were observed in 100,000 WBM cells. We found 289 metabolites that were detected in 100,000 WBM cells. This 289 metabolite library was used for manual analyses of LC-MS data from low abundance samples. When additional metabolites were observed in new experiments they were added to the library.

## Melanoma specimens

Melanoma specimens were obtained with informed consent from all patients according to protocols approved by the Institutional Review Board (IRB) of the University of Michigan Medical School (IRBMED approvals HUM00050754 and HUM00050085 *Quintana et al., 2012*) and the University of Texas Southwestern Medical Center (IRB approval 102010–051). Materials used in the manuscript

are available, though there are restrictions imposed by IRB requirements and institutional policy on the sharing of materials from patients.

## Mouse studies and xenograft assays

All mouse experiments complied with all relevant ethical regulations and were performed according to protocols approved by the Institutional Animal Care and Use Committee at the University of Texas Southwestern Medical Center (protocols 2016–101360 and 2019–102632). For all experiments, mice were kept on normal chow and fed ad libitum. The mice used in all experiments were 8–12-week-old C57BL/Ka mice, with the exception of melanoma studies, which were subcutaneously xenografted into 4–8-week-old NOD.CB17-*Prkdc^{scid} Il2rg^{tm1Wjl}*/SzJ (NSG) mice. Both male and female mice were used. For melanoma experiments, the maximum permitted tumor diameter was 2.5 cm. Subcutaneous tumor diameters were measured weekly with calipers until any tumor in the mouse cohort reached 2.5 cm in its largest diameter. At that point, all mice in the cohort were killed, per approved protocol, for analysis of subcutaneous tumors and circulating melanoma cells. For each replicate, subcutaneous tumors and circulating melanoma cells were pooled from 6 to 10 mice.

## Methotrexate treatment

Eight- to twelve-week old C57BL/Ka mice were intraperitoneally injected daily with methotrexate (1.25 mg/kg/day) or DMSO control, for 3 days. Mice were sacrificed by cervical dislocation 2 hr after the final methotrexate dose and bone marrow cells were collected for analysis.

## Statistical analysis of metabolomic data

We developed an R tool for the analysis of metabolite LC-MS peak intensity data. The data were visualized using multiple methods, including violin-box plots, histograms, clustered heatmaps, principle component analysis, and correlation plots to assess data quality and identify batch effects. To assess the statistical significance of differences in metabolite levels between samples we used R's generalized linear models (GLM) (*Dobson and Barnett, 2018*) function with the Gaussian distribution on log-transformed data. To compare metabolite levels in cell samples to sheath fluid samples we used GLM with $\log_2(x+1)$-transformed, non-normalized data. Metabolites with fold change >2 and FDR < 0.05 were considered above background. To assess the statistical significance of differences in metabolite levels between two types of cells, we normalized the cell samples using the relative log expression (RLE) method (*Anders and Huber, 2010*), and $\log_2$-transformed the normalized data. For all comparisons between samples, we used the half-minimum imputation to replace zero values with half of the minimum nonzero value for each metabolite, and used R's GLM method. To adjust for multiple comparisons we used the FDR method. For comparisons between different cell samples, we used fold change >2 and FDR < 0.05 as cutoffs for statistical significance. When samples (such as HSCs and WBM cells) were from the same mice we used pairing as an independent variable in the GLM. When batch effects were observed, we used batch as an independent variable in the GLM. R packages used by this tool include stats, openxlsx, data.table, gtools, matrixStats, cplm, ggplot2, cowplot, pheatmap, ggcorrplot, eulerr, and GGally.

The metabolomics data analysis tool can be downloaded from https://git.biohpc.swmed.edu/CRI/ODA for academic use. This tool includes an ODA.R script file, an accompanying Excel data template file, and example analyses. The script can be run from Linux/MacIntosh Terminal or Windows PowerShell using the Rscript command followed by the Excel input file name and the Excel output file name. R with the Rscript command (version 3.5.1 or later is recommended) and internet access are required to run this tool as other R packages must be auto-downloaded by the tool. Data should be entered into the Excel template and parameters for analysis selected. First-time users should read the instructions in the data template. The analysis reports and figures are saved together in the Excel output file. Figures are also saved in a folder in the .png and .ps formats. Example analysis results are provided to illustrate typical analysis settings and their outputs.

## Assessing statistical significance

Mice were allocated to experiments randomly and samples processed in an arbitrary order, but formal randomization techniques were not used. Prior to analyzing the statistical significance of differences among treatments, we tested whether the data were normally distributed and whether

variance was similar among treatments. To test for normal distribution, we performed the Shapiro–Wilk test when $3 \leq n < 20$ or the D'Agostino Omnibus test when $n \geq 20$. To test if variability significantly differed among treatments, we performed $F$-tests (for experiments with two treatments) or Levene's median tests (for more than two treatments). When the data significantly deviated from normality or variability significantly differed among treatments, we log2-transformed the data and tested again for normality and variability. If the transformed data did not significantly deviate from normality and equal variability, we performed parametric tests on the transformed data. Fold change data were always log2-transformed.

All the statistical tests we used were two-sided, where applicable. To assess the statistical significance of a difference between two treatments, we used Student's t-tests or paired t-tests (when a parametric test was appropriate). To assess the statistical significance of differences between two cumulative frequency distributions, we used the Kolmogorov–Smirnov tests. Multiple Kolmogorov–Smirnov tests were followed by Holm–Sidak's multiple comparisons adjustment. To assess the statistical significance of differences between more than two treatments, we used paired sample one-way or two-way ANOVAs (when a parametric test was appropriate) followed by Tukey's, Dunnet's, or Sidak's multiple comparisons adjustment. To assess the statistical significance of differences between transplant data, we used mixed-effects analysis (when a parametric test was appropriate and there were missing data points) followed by Sidak's multiple comparisons adjustment. To assess the correlation between two sets of samples, we calculated Spearman correlation coefficients (r, the data were not normally distributed) and performed linear regression analysis.

All statistical analyses were performed with Graphpad Prism 8.3. All data represent mean ± standard deviation. Samples sizes were not pre-determined based on statistical power calculations but were based on our experience with these assays. No data were excluded; however, mice sometimes died during experiments, presumably due to complications associated with irradiation and bone marrow transplantation. In those instances, data that had already been collected on the mice in interim analyses were included (such as donor contribution to peripheral blood chimerism over time).

## Acknowledgements

SJM is a Howard Hughes Medical Institute (HHMI) Investigator, the Mary McDermott Cook Chair in Pediatric Genetics, the Kathryn and Gene Bishop Distinguished Chair in Pediatric Research, the director of the Hamon Laboratory for Stem Cells and Cancer, and a Cancer Prevention and Research Institute of Texas Scholar. AWD was supported by a Ruth L Kirschstein NRSA Fellowship. AT was supported by the Leopoldina Fellowship Program (LPDS 2016–16) of the German National Academy of Sciences and the Fritz Thyssen Foundation. This work was also funded by the National Institutes of Health (DK11875 and CA228608) and the Cancer Prevention and Research Institute of Texas (RP180778). Jian Xu provided the AML and ALL cell lines used in this study. The BioHPC high performance computing cloud at UTSW was used for data analysis and storage as well as metabolomics data analysis (ODA) software deployment. The Moody Foundation Flow Cytometry Core was used for all flow cytometry. The graphic in *Figure 4B* was created using BioRender (BioRender.com).

## Additional information

### Competing interests

Sean J Morrison: advisor for Frequency Therapeutics and Protein Fluidics as well as a stockholder in G1 Therapeutics. The other authors declare that no competing interests exist.

### Funding

| Funder | Grant reference number | Author |
|---|---|---|
| Howard Hughes Medical Institute | | Thomas P Mathews<br>Sean J Morrison |
| National Institutes of Health | | Sean J Morrison |
| National Institutes of Health | DK11875 | Sean J Morrison |

| National Institutes of Health | CA228608 | Sean J Morrison |
| Cancer Prevention and Research Institute of Texas | RP180778 | Sean J Morrison |
| National Institutes of Health | F32 HL 135975 | Andrew W DeVilbiss |
| Fritz Thyssen Foundation | | Alpaslan Tasdogan |
| German National Academy of Sciences Leopoldina Fellowship Program | LPDS 2016-16 | Alpaslan Tasdogan |

The funders had no role in study design, data collection and interpretation, or the decision to submit the work for publication.

### Author contributions

Andrew W DeVilbiss, Thomas P Mathews, Conceptualization, Data curation, Formal analysis, Investigation, Methodology, Writing - original draft, Writing - review and editing; Zhiyu Zhao, Data curation, Software, Formal analysis; Misty S Martin-Sandoval, Data curation, Formal analysis, Investigation; Jessalyn M Ubellacker, Investigation; Alpaslan Tasdogan, Formal analysis, Investigation; Michalis Agathocleous, Conceptualization, Formal analysis, Supervision, Methodology, Writing - review and editing; Sean J Morrison, Conceptualization, Resources, Formal analysis, Supervision, Funding acquisition, Project administration, Writing - review and editing

### Author ORCIDs

Andrew W DeVilbiss (iD) https://orcid.org/0000-0002-9739-2543
Zhiyu Zhao (iD) http://orcid.org/0000-0001-6308-6997
Thomas P Mathews (iD) https://orcid.org/0000-0002-9355-8243
Sean J Morrison (iD) https://orcid.org/0000-0003-1587-8329

### Ethics

Animal experimentation: All mouse experiments complied with all relevant ethical regulations and were performed according to protocols approved by the Institutional Animal Care and Use Committee at the University of Texas Southwestern Medical Center (protocols 2016-101360 and 2019-102632).

### Decision letter and Author response

Decision letter https://doi.org/10.7554/eLife.61980.sa1
Author response https://doi.org/10.7554/eLife.61980.sa2

## Additional files

### Supplementary files

• Transparent reporting form

### Data availability

All data generated or analyzed during this study are included in the manuscript and the source data files.

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
