## [Decision Letter]

**Acceptance summary:**

In this manuscript, the authors described mass spectrometry methods for measuring metabolites in small numbers of cells. They also use methods to measure metabolite changes after sorting, and use these methods to measure metabolite changes after methotrexate treatment or between subcutaneous melanoma cells and circulating tumor cells. While the time required to sort cells is fast relative to some metabolic processes, these caveats are discussed and this work demonstrates that some insight can be gained from metabolomics analysis of rare cell populations.

**Decision letter after peer review:**

Thank you for submitting your article "Metabolomic profiling of rare cell populations isolated by flow cytometry from tissues" for consideration by *eLife*. Your article has been reviewed by three peer reviewers, one of whom is a member of our Board of Reviewing Editors, and the evaluation has been overseen by Richard White as the Senior Editor. The following individual involved in review of your submission has agreed to reveal their identity: Thales Papagiannakopoulos (Reviewer #3).

The reviewers have discussed the reviews with one another and the Reviewing Editor has drafted this decision to help you prepare a revised submission.

In this paper the authors report methods to measure metabolites in a small numbers of cell. This work has broad applications for metabolite analysis in rare cell populations and will be a good resource for the field.

Overall, the three reviewers found the work to be of interest and well done, but felt some additional controls and/or discussion be included prior to publication. We are providing a summary of the 6 points below we feel need to be fully addressed. In addition, please address the technical concerns raised by reviewer #2 (below). We are also providing the other full reviewer #1/#3 comments for your reference, which are not required for publication but if you have any additional text comments or data those would be welcome.

Essential revisions:

1) Please provide more details on how the sorting versus pipetting control was performed, and specify which metabolites were most sensitive to sorting.

2) A comparison of sorted and un-sorted cells (a rapidly quenched sample) would be useful to include, again with those metabolites that are most sensitive to sorting specified. At a minimum, the authors should acknowledge the need for this control if new data cannot be provided.

3) Consider providing a table listing the metabolites that significantly differed in sorted populations relative to various controls

4) Please specify how the minimum number of cells needed for analysis was determined?

5) Please discuss whether factors such as hypo-osmolarity or addition of glucose to the sorting buffer may affect metabolite measurements, and if any data are available to address how this affects adding it may be useful for the field..

6) Please clarify if internal standards were used to quantify metabolites, and address any other technical questions and concerns revised by reviewer #2.

Reviewer #1:

In this manuscript, the authors described mass spectrometry methods for measuring metabolites in small numbers of cells. They also use methods to measure metabolite changes after sorting, and use these methods to measure metabolite changes after methotrexate treatment or between subcutaneous melanoma cells and circulating tumor cells. The work is well done and should be a valuable resource for field, although potential caveats to consider when applying the method could be better discussed to help guide its use by other groups.

1) It is important that the authors tested variables like extraction solvent and drying techniques in their system, although it would be appropriate to also cite the published literature describing which metabolite classes are enriched in different extraction solvents and the effects of different drying methods.

2) In Figure 1H-J, the authors investigate metabolite changes from sorting and found that most metabolite differences between unsorted cells were also found from comparing sorted cells. They conclude that sorting does not affect detection of most metabolite differences, however there was still some effect of sorting, particularly for the AML cells, as shown by PCA in Figure 1H. It could be helpful to more explicitly state which metabolites are sensitive to sorting.

3) It is surprising that most metabolite levels do not over time with cell isolation, although it is worth pointing out that the comparison is to a time point 5 minutes after harvesting. Some might argue that a comparison to a rapidly quenched sample is needed too, although practical considerations may make this difficult for the cell types considered in the study. Nevertheless, some discussion of this point could be helpful as some in the field argue it is necessary to go to great lengths to quench metabolism for assessment of some rapidly changing processes.

4) The metabolites that change from Figure 1—figure supplement 1 appear to be largely nucleotide or lipid related metabolites. Did amino acids and glucose-derived metabolites such as pyruvate and lactate also change over this time course? It might be expected that levels of some would change as levels are known to be sensitive to exchange with the media and cells were maintained in HBSS after being flushed from bone marrow. Since HBSS contains glucose, this buffer may help stabilized some metabolite levels over the conditions compared, but what is included in this buffer may also influence what is measured and is stable over time. Thus, what solution cells are isolated in is thus another variable that could be considered when applying this technique.

5) Another point to consider is that the cutoff used for counting a metabolite level as enriched or depleted in Figure 1 is a fold change of 2 or greater. However, in some cases a <2 fold change in metabolite levels may be significant depending on the question considered as another issue to consider when applying this method.

Reviewer #2:

The majority of metabolomic measurements are carried out on bulk populations of cells that do not reflect the heterogeneity of biological tissues. The Morrison lab previously published methods for isolating metabolites from abundant sorted cells (Agathocleous et al., 2017). Here, the authors use HILIC chromatography and a high resolution Orbitrap MS to detect a larger number of metabolites in smaller numbers of cells. The authors isolate HSCs and circulating melanoma cells using their method, and demonstrate that expected metabolites such as AICAR increase in HSCs following methotrexate treatment.

The method will be useful and of interest to the general community if the authors can address the following technical queries:

1) The sheath fluid, 0.5x PBS, is hypotonic. The authors should comment on the effects of hypoosmotic stress on these cells and ideally provide EM or dye exclusion (e.g. π or trypan blue) data to verify the integrity of cells incubated in 0.5% PBS for the duration of the sort.

2) The authors verified the identities of their metabolites with chemical standards. To the best of this reviewer's understanding, though, the differences in metabolites are based on changes in counts. There were no standard curves run with a concentration range of these small molecules to demonstrate that these counts were within the dynamic range of the instrument. I also could not determine if LC-MS performance was assessed with internal standards included in the extraction buffer. Please correct me if I am wrong on both of these.

If there are no internal standards, then the authors should include, at a minimum, a set of isotopically labeled internal standards or an unnatural amino acid such as norvaline in the extraction solvent to which all counts should be normalized to account for changes in instrument performance. Standard curves should also be run for the metabolites that change to verify that the changes in metabolite abundance are within the dynamic range of the MS detector and that the counts are neither too low nor saturating.

3) Along the same lines, the non-polar metabolites that elute at the beginning of a HILIC run often partition poorly onto the column. The authors should confirm that these metabolites, or at least a subset (given that there are many lipid species) have good peak shapes and widths indicative of chromatographic resolution of these metabolites (at least 5 points on either side of the peak; see Kadjo et al., Anal. Chem. 2017, 89, 3884−3892) and provide standard curves for at least the classes of these metabolites to show that changes in standard concentrations of these small molecules are reflected in the peaks and counts that they see.

4) In Figure 1I, the authors compare sorted and AML and ALL cell metabolite abundances and find that most of the metabolites that differed between the cell lines also differed between when the same cell line was sorted or pipetted. These data are very important as they are critical to the utility of the method. The authors use a cutoff >2-fold change with an FDR of <0.01 as a cutoff for significant differences, but a 2-fold change in many metabolites, including NADPH (which is one of the metabolites that changes significantly) is quite important. Could the authors provide the counts and fold changes for all detected metabolites in pipetted and sorted cells, and also describe how many metabolites are significantly changed if the cutoffs are less stringent (for example, 1.5-fold with an FDR of 0.05)? This will give the community a good idea of the metabolites that are stable following sorting.

5) In Figures 1O and P, the abundances of many of the metabolites, including aspartate and malate, are in the 1E8-1E9 range, which are very high for an Orbitrap, particularly for 10000 cells. This may be due to the Orbitrap AGC settings (1E6 target ion counts; max inject time of 50 ms). Could the authors include details on how they optimized their AGC settings and how often this target count is reached?

Reviewer #3:

DeVilbiss et al. describe an improved method to analyze metabolites of rare cell populations in vivo. The authors adjusted their previously described techniques to increase the overall number of metabolites detected and to reduce contaminants from sample prep.

The technique described in this manuscript enables the authors to reliably detect 160 metabolites, more than 2.5X the number of metabolites using previous method, from a population of 10,000 sorted cells. The method is robust with the majority of metabolites showing strong correlation between sorted and unsorted samples and only a minority of metabolites differing significantly over time periods necessary for sample prep and processing. Overall this manuscript describes a valuable new method to interrogate metabolic changes in rare cell populations in vivo that will further the field.

With a few revisions this manuscript is suitable for publication in *eLife*.

1) How was the minimum number of cells necessary determined (10,000)? What is the minimum number of cells necessary to identify a similar number of metabolites reliably? While this represents a significant improvement to previous methods, pooling of material would still be required in certain contexts. To acquire 10,000 circulating melanoma cells for analysis in Figure 3, the authors pooled 6-10 mice for analysis.

2) Are certain metabolites/classes of metabolites more robust during the sorting process? Please provide a table related to Figure 1I listing the metabolites that significantly differed between sorting vs. pipetting.

---

## [Author Response]

Essential revisions:1) Please provide more details on how the sorting versus pipetting control was performed, and specify which metabolites were most sensitive to sorting.

We have updated the Materials and methods section to include a more detailed description of how this experiment was performed. Non-adherent cultures of AML or ALL cells were removed from the incubator and centrifuged at 4°C to pellet the cells, then the cells were resuspended in ice-cold PBS and centrifuged again to wash, before being resuspended in ice cold PBS at a density of 1x10^6^ cells/ml. 10 µl of this AML or ALL cell suspension was then pipetted into 40 µl of 100% acetonitrile to create a final extract of 10,000 cells in 50 µl 80% acetonitrile. This was all done in less than 10 minutes after removing the cells from the incubator. In the other arm of the experiment, 10,000 AML or ALL cells from the same cell suspensions were sorted into 40 µl of 100% acetonitrile by flow cytometry to extract metabolites from 10,000 cells in 50 µl of 80% acetonitrile. Sorting of these samples was completed within 30 to 60 minutes after removing the cultures from the incubator. The cell suspensions were always kept ice cold before and during sorting. As detailed in points 2 and 3 below, we have added source data files to the revised manuscript listing which metabolites differed between sorted and unsorted samples.

2) A comparison of sorted and un-sorted cells (a rapidly quenched sample) would be useful to include, again with those metabolites that are most sensitive to sorting specified. At a minimum, the authors should acknowledge the need for this control if new data cannot be provided.

The experiment described in response to point #1 compared sorted and unsorted cells. The unsorted cells were quenched in acetonitrile less than 10 minutes after removal from the incubator, whereas the sorted cells were quenched in 30-60 minutes. In pipetted samples, we detected 97 metabolites that significantly differed between AML and ALL cells (fold change > 2.0 and FDR < 0.05) (Figure 1I). In sorted samples, we detected 85 metabolites that significantly differed between AML and ALL cells. Of these, 71 significantly differed among AML and ALL cells in both sorted and pipetted samples. Figure 1—source data 3 lists all of the metabolites that significantly differed between sorted and pipetted AML cells. Figure 1—source data 4 lists all of the metabolites that significantly differed between sorted and pipetted ALL cells. Figure 1—source data 5 lists all of the metabolites that significantly differed between AML and ALL cells in both sorted and pipetted samples, or only in the sorted samples, or only in the pipetted samples.

3) Consider providing a table listing the metabolites that significantly differed in sorted populations relative to various controls

We have provided 6 source data files (5 of them new) listing metabolites that differed among various conditions.

– Related to Figure 1F, we list the metabolites that significantly differed among cells sorted with 1x PBS sheath fluid versus 0.5x PBS sheath fluid (new Figure 1—source data 2)

– Related to Figure 1H, we list the metabolites that significantly differed between sorted and pipetted AML cells (new Figure 1—source data 3) and between sorted and pipetted ALL cells (new Figure 1—source data 4)

– Related to Figure 1I, we list the metabolites that significantly differed between AML and ALL cells in sorted and pipetted samples, only in the sorted samples, and only in the pipetted samples (new Figure 1—source data 5)

– Related to Figures 1L-P, we list the metabolites that differed between cells that were incubated on ice for increasing amounts of time (Figure 1—source data 6).

– We provided new data identifying the metabolites that significantly differed between whole bone marrow cells suspended in HBSS versus PBS (Figure 1—source data 7).

4) Please specify how the minimum number of cells needed for analysis was determined?

It’s not that there is a minimum number of cells for analysis, it’s just that the number of metabolites detected declines as cell number declines. So we try to analyze as many cells as can readily be isolated by flow cytometry. For rare cell populations, such as HSCs (0.007% of whole bone marrow cells) it is difficult to routinely isolate more than 10,000 cells in a single experiment. Consequently, we performed many of our control experiments to evaluate the quality of the metabolomic data using 10,000 sorted cells. We compared the numbers of metabolites detected above background in aliquots of 10,000 to 100,000 cells in Figure 1G.

5) Please discuss whether factors such as hypo-osmolarity or addition of glucose to the sorting buffer may affect metabolite measurements, and if any data are available to address how this affects adding it may be useful for the field..

The 0.5x PBS sheath fluid is not likely to cause hypotonic stress during sorting because the cells were suspended in 1x HBSS and there is laminar flow within the cytometer such that there is limited mixing of the sample buffer and the sheath fluid. Moreover, cells are exposed to sheath fluid for less than a second as they pass through the flow cytometer, leaving little time for mixing or for hypotonic stress responses before the cells are lysed in acetonitrile.

To address this, we compared metabolomic data from 10,000 whole bone marrow cells sorted with 0.5x PBS versus 1x PBS sheath fluid in Figure 1F. Using 0.5x PBS we detected 157±2 metabolites above background and using 1x PBS we detected 149±8 metabolites above background. 18 metabolites significantly differed between samples sorted with 0.5x versus 1x PBS sheath fluid (see new Figure 1—source data 2). Apart from potential effects of hypo-osmolarity, these differences also reflect differences in ion suppression or metabolite extraction efficiency as a result of the differences in salt concentrations in the extraction buffer. We have updated the text related to Figure 1F to reflect these new data.

To assess the effect of glucose in the sample buffer, we compared the metabolomic profiles of 10,000 bone marrow cells sorted from samples suspended in either HBSS (which contains glucose) or PBS (which does not). When cells were suspended in HBSS we detected 162 metabolites above background and when suspended in PBS we detected 150 metabolites above background. Only 5 metabolites significantly differed between the HBSS versus PBS samples (Figure 1—source data 7). As expected, glucose levels were substantially higher in the cells that had been suspended in HBSS.

6) Please clarify if internal standards were used to quantify metabolites, and address any other technical questions and concerns revised by reviewer #2.

Absolute metabolite concentrations can be measured using isotopically-labeled internal standards in methods designed to quantitate small numbers of metabolites but this is impractical with metabolomics methods designed to measure hundreds of different metabolites. Like other omics analyses, metabolomics is a hypothesis generating tool that broadly surveys the relative levels of many metabolites. We routinely follow-up metabolomics analyses with other methods optimized to extract and quantitate particular species. In those follow-up studies we include labelled internal standards to determine absolute concentrations (e.g. Nature 577:115, Nature 527:186, Nature 585:113). We have noted this in the Discussion.

To test whether the metabolites we analyzed fell within the linear range of the mass spectrometer, we added a mixture of four isotopically labeled lipids, corresponding to the four major lipid classes that we detected, at successive dilutions to extracts of 10,000 bone marrow cells. This allowed us to generate standard curves for these lipids in the context of the matrix effects from the cell extract. We included data in the revised manuscript showing the standard curves for each of these lipid classes including phosphatidylethanolamine (PE), phosphatidylcholine (PC), lysophosphatidylethanolamine (Lyso-PE), and lysophosphatidylcholine (Lyso-PC) (Figure 1—figure supplement 1E-H). All regression analyses were highly linear (R^2^ > 0.95) and lipids from these classes extracted from 10,000 bone marrow cells fell within the linear range.

Other questions from each of the reviewers are addressed below.

Reviewer #1:In this manuscript, the authors described mass spectrometry methods for measuring metabolites in small numbers of cells. They also use methods to measure metabolite changes after sorting, and use these methods to measure metabolite changes after methotrexate treatment or between subcutaneous melanoma cells and circulating tumor cells. The work is well done and should be a valuable resource for field, although potential caveats to consider when applying the method could be better discussed to help guide its use by other groups.1) It is important that the authors tested variables like extraction solvent and drying techniques in their system, although it would be appropriate to also cite the published literature describing which metabolite classes are enriched in different extraction solvents and the effects of different drying methods.

In the original manuscript we cited a study that found that acetonitrile-based extraction is superior to methanol extraction for certain classes of metabolites (Analytical Chemistry 79:6167) and a study that showed that drying accelerates the decomposition of some metabolites (Nature 510:298). We have addressed this issue in more depth in the revised manuscript by adding a new paragraph to the Discussion that cites additional papers that address the advantages and disadvantages of different extraction solvents and drying methods.

2) In Figure 1H-J, the authors investigate metabolite changes from sorting and found that most metabolite differences between unsorted cells were also found from comparing sorted cells. They conclude that sorting does not affect detection of most metabolite differences, however there was still some effect of sorting, particularly for the AML cells, as shown by PCA in Figure 1H. It could be helpful to more explicitly state which metabolites are sensitive to sorting.

We have added files listing the metabolites that significantly differed between sorted AML cells versus pipetted AML cells (Figure 1—source data 3), and sorted ALL cells versus pipetted ALL cells (Figure 1—source data 4).

3) It is surprising that most metabolite levels do not over time with cell isolation, although it is worth pointing out that the comparison is to a time point 5 minutes after harvesting. Some might argue that a comparison to a rapidly quenched sample is needed too, although practical considerations may make this difficult for the cell types considered in the study. Nevertheless, some discussion of this point could be helpful as some in the field argue it is necessary to go to great lengths to quench metabolism for assessment of some rapidly changing processes.

It is difficult to address this issue in the context of our study because 5 minutes is already very fast for getting bone marrow cells out of a mouse. Even getting cells growing in suspension out of a culture dish, and centrifuging to eliminate the culture medium would take a few minutes. Therefore, in the context of a protocol that involves the isolation of cells by flow cytometry it is hard to test the idea that some metabolites may change in less than a minute after cells leave their normal physiological environment. Nonetheless, as noted by the reviewer this issue has previously been addressed by other studies that did not examine sorted cells (Annu Rev Biochem. 20:277). We have cited this review in the Discussion of our paper and noted that some metabolites may change in less than a minute after leaving their normal physiological environment in a way that would not have been detected in our experiments.

4) The metabolites that change from Figure 1—figure supplement 1 appear to be largely nucleotide or lipid related metabolites. Did amino acids and glucose-derived metabolites such as pyruvate and lactate also change over this time course? It might be expected that levels of some would change as levels are known to be sensitive to exchange with the media and cells were maintained in HBSS after being flushed from bone marrow.

Figure 1—figure supplement 1 is now Figure 1—source data 6 in the revised manuscript. We did not observe changes in amino acid levels over time in cell suspensions that incubated on ice and we did not detect pyruvate in any of these samples so do not know whether it changed. Lactate did not significantly change at most time points, including after 4 hours of incubation, but was significantly reduced at 3 hours as compared to 5 minutes.

Since HBSS contains glucose, this buffer may help stabilized some metabolite levels over the conditions compared, but what is included in this buffer may also influence what is measured and is stable over time. Thus, what solution cells are isolated in is thus another variable that could be considered when applying this technique.

We compared the metabolic profiles of 10,000 bone marrow cells sorted from samples suspended in either HBSS or PBS. When cells were suspended in HBSS we detected

162 metabolites above background and in PBS we detected 150 metabolites above background. Only 5 metabolites significantly differed between the HBSS versus PBS samples (Figure 1—source data 7). As expected, glucose levels were substantially higher in the cells that had been suspended in HBSS.

5) Another point to consider is that the cutoff used for counting a metabolite level as enriched or depleted in Figure 1 is a fold change of 2 or greater. However, in some cases a <2 fold change in metabolite levels may be significant depending on the question considered as another issue to consider when applying this method.

We agree that any fold change cutoff is arbitrary and that there are advantages and disadvantages to raising or lowering it. It is true that some changes less than 2-fold could be biologically important. On the other hand, changes less than 2-fold are less likely to be confirmed in follow up experiments and we wanted to use a reasonably robust threshold when determining the number of metabolites detected above background. To balance these considerations, and in an effort to use common thresholds throughout the paper, we have consistently identified metabolites that differed among samples based on fold change > 2 and FDR < 0.05. To address the concern that some metabolites with fold change < 2 might be biologically significant we have added additional lists of metabolites to Figure 1—source data 2-7 that show metabolites that were fold change = 1.5 to 2.0 and FDR < 0.05.

Reviewer #2:[…] The method will be useful and of interest to the general community if the authors can address the following technical queries:1) The sheath fluid, 0.5x PBS, is hypotonic. The authors should comment on the effects of hypoosmotic stress on these cells and ideally provide EM or dye exclusion (e.g. π or trypan blue) data to verify the integrity of cells incubated in 0.5% PBS for the duration of the sort.

The 0.5x PBS sheath fluid is not likely to cause hypotonic stress during sorting because the cells were suspended in 1x HBSS and there is laminar flow within the cytometer such that there is limited mixing of the sample buffer and the sheath fluid. Moreover, cells are exposed to sheath fluid for less than a second as they pass through the flow cytometer, leaving little time for mixing or for hypotonic stress responses before the cells are lysed in acetonitrile.

To test if the cells experience hypotonic stress during sorting we performed a dye exclusion assay in which we included the viability dye DAPI in a bone marrow cell suspension, then analyzed the cells on a flow cytometer with 0.5x or 1.0x PBS sheath fluid. We did not observe any difference in the intensity of DAPI staining or the percentage of DAPI^+^ cells among samples:

2) The authors verified the identities of their metabolites with chemical standards. To the best of this reviewer's understanding, though, the differences in metabolites are based on changes in counts. There were no standard curves run with a concentration range of these small molecules to demonstrate that these counts were within the dynamic range of the instrument. I also could not determine if LC-MS performance was assessed with internal standards included in the extraction buffer. Please correct me if I am wrong on both of these.

Absolute metabolite concentrations can be measured using isotopically-labeled internal standards in methods designed to quantitate small numbers of metabolites but this is impractical with metabolomics methods designed to measure hundreds of metabolites. Like other omics analyses, metabolomics is a hypothesis generating tool that broadly surveys the relative levels of many metabolites between samples. We routinely follow-up metabolomics analyses with other methods optimized to extract and quantitate particular species. In those follow-up studies we include labelled internal standards to determine absolute concentrations (e.g. Nature 577:115, Nature 527:186, Nature 585:113). We have noted this in the Discussion.

Instrument performance was evaluated and normalized across runs and among samples by comparing average signal intensity for all metabolites detected above background. After peak integration from raw data files the individual peak areas of all metabolites were log transformed and averaged. Each log transformed metabolite was then normalized to the average of all log transformed peaks within a sample such that average signal intensity was the same across samples. This allowed us to detect and control for changes in instrument performance.

Instrument performance was evaluated before each experiment by analyzing a quality control sample (20ul of freshly-obtained rat serum). We compared peak areas of individual metabolites and the total number of metabolites detected in the sample with control samples run in prior experiments. If peak area values and total metabolite identifications fell outside of 1 standard deviation from the historical average, the instrument was cleaned to re-optimize sensitivity.

If there are no internal standards, then the authors should include, at a minimum, a set of isotopically labeled internal standards or an unnatural amino acid such as norvaline in the extraction solvent to which all counts should be normalized to account for changes in instrument performance. Standard curves should also be run for the metabolites that change to verify that the changes in metabolite abundance are within the dynamic range of the MS detector and that the counts are neither too low nor saturating.

To test if the metabolites we analyzed fell within the linear range of the mass spectrometer we added a mixture of four isotopically labeled lipids corresponding to the four major lipid classes that we detected at successive dilutions to extracts of 10,000 bone marrow cells. This allowed us to generate standard curves for these lipids in the context of the matrix effects from the cell extract. We included data showing the standard curves for each of these lipid classes including phosphatidylethanolamine (PE), phosphatidylcholine (PC), lysophosphatidylethanolamine (Lyso-PE), and lysophosphatidylcholine (Lyso-PC) (Figure 1—figure supplement 1 E-H). All regression analyses were highly linear (R^2^ > 0.95) and lipids from these classes extracted from 10,000 bone marrow cells fell within the linear range.

3) Along the same lines, the non-polar metabolites that elute at the beginning of a HILIC run often partition poorly onto the column. The authors should confirm that these metabolites, or at least a subset (given that there are many lipid species) have good peak shapes and widths indicative of chromatographic resolution of these metabolites (at least 5 points on either side of the peak; see Kadjo et al., Anal. Chem. 2017, 89, 3884−3892) and provide standard curves for at least the classes of these metabolites to show that changes in standard concentrations of these small molecules are reflected in the peaks and counts that they see.

To address this we added graphs of chromatographic peaks of four lipid species that we detected in extracts of 10,000 bone marrow cells, representing the major lipid classes including a PC, a PE, a Lyso-PC, and a Lyso-PE (Figure 1—figure supplement 1A-D). Each lipid species exhibited good peak shape, with at least 8 scans across each peak. Figure 1—figure supplement 1E-H shows that changes in the concentrations of isotopically-labelled internal standards for each of these lipid classes were reflected by changes in peak intensity, with highly linear correlations and R^2^ values > 0.95.

4) In Figure 1I, the authors compare sorted and AML and ALL cell metabolite abundances and find that most of the metabolites that differed between the cell lines also differed between when the same cell line was sorted or pipetted. These data are very important as they are critical to the utility of the method. The authors use a cutoff >2-fold change with an FDR of <0.01 as a cutoff for significant differences, but a 2-fold change in many metabolites, including NADPH (which is one of the metabolites that changes significantly) is quite important. Could the authors provide the counts and fold changes for all detected metabolites in pipetted and sorted cells, and also describe how many metabolites are significantly changed if the cutoffs are less stringent (for example, 1.5-fold with an FDR of 0.05)? This will give the community a good idea of the metabolites that are stable following sorting.

We agree that any fold change cutoff is arbitrary and that there are advantages and disadvantages to raising or lowering it. It is true that some changes less than 2-fold could be biologically important. On the other hand, changes less than 2-fold are less likely to be confirmed in follow up experiments and we wanted to use a reasonably robust threshold when determining the number of metabolites detected above background. To balance these considerations, and in an effort to use common thresholds throughout the paper, we have consistently identified metabolites that differed among samples based on fold change > 2 and FDR < 0.05. To address the concern that some metabolites with fold change < 2 might be biologically significant we have added additional lists of metabolites to Figure 1—source data 2-7, that show metabolites that were fold change = 1.5 to 2.0 and FDR < 0.05.

5) In Figures 1O and P, the abundances of many of the metabolites, including aspartate and malate, are in the 1E8-1E9 range, which are very high for an Orbitrap, particularly for 10000 cells. This may be due to the Orbitrap AGC settings (1E6 target ion counts; max inject time of 50 ms). Could the authors include details on how they optimized their AGC settings and how often this target count is reached?

The ACG settings we used were as recommended by Thermo Scientific. A study conducted by Thermo Scientific found that an AGC target of 1E6 with a 50ms trap fill time are ideal trap fill parameters on Q-Exactive instruments (Anal. Chem. 86:116). These parameters were also used in our prior studies (Nature 577:115, Nature 585:113). When ions are being introduced into the source, our trap fills with 1E6 ions in 2-5 ms (depending on the exact point in the chromatogram) and the target count is reached on every scan.

Reviewer #3:[…] With a few revisions this manuscript is suitable for publication in eLife.1) How was the minimum number of cells necessary determined (10,000)? What is the minimum number of cells necessary to identify a similar number of metabolites reliably? While this represents a significant improvement to previous methods, pooling of material would still be required in certain contexts. To acquire 10,000 circulating melanoma cells for analysis in Figure 3, the authors pooled 6-10 mice for analysis.

It’s not that there is a minimum number of cells for analysis, it’s just that the number of metabolites detected declines as cell number declines. We try to analyze as many cells as can readily be isolated by flow cytometry. For rare cell populations, such as HSCs (0.007% of whole bone marrow cells) it is difficult to routinely isolate more than 10,000 cells in a single experiment. Consequently, we performed many of our control experiments to evaluate the quality of the metabolomic data using 10,000 sorted cells. The method can be used to assay fewer cells, with the detection of fewer metabolites but the exact numbers would depend on the cell type. We compared the numbers of metabolites detected above background in aliquots of 10,000 to 100,000 cells in Figure 1G and detected only modestly higher numbers of metabolites in 100,000 cells (222 metabolites) as compared to 10,000 cells (157 metabolites).

2) Are certain metabolites/classes of metabolites more robust during the sorting process? Please provide a table related to Figure 1I listing the metabolites that significantly differed between sorting vs. pipetting.

As detailed in response to point #3 of the Essential revisions, we added a number of source data files listing the metabolites that differed in several different experiments. Related to Figure 1I, we list the metabolites that significantly differed between AML and ALL cells in sorted and pipetted samples, only in the sorted samples, and only in the pipetted samples (Figure 1—source data 5).